# Bioinformatic and cell-based tools for pooled CRISPR knockout screening in mosquitos

Raghuvir Viswanatha[1,4 ✉], Enzo Mameli [1,2,4], Jonathan Rodiger[1], Pierre Merckaert[1],
Fabiana Feitosa-Suntheimer[2], Tonya M. Colpitts[2], Stephanie E. Mohr [1], Yanhui Hu [1] &
Norbert Perrimon [1,3 ✉]

Mosquito-borne diseases present a worldwide public health burden. Current efforts to understand and counteract them have been aided by the use of cultured mosquito cells. Moreover, application in mammalian cells of forward genetic approaches such as CRISPR screens have identified essential genes and genes required for host-pathogen interactions, and in general, aided in functional annotation of genes. An equivalent approach for genetic screening of mosquito cell lines has been lacking. To develop such an approach, we design a new bioinformatic portal for sgRNA library design in several mosquito genomes, engineer mosquito cell lines to express Cas9 and accept sgRNA at scale, and identify optimal promoters for sgRNA expression in several mosquito species. We then optimize a recombination-mediated cassette exchange system to deliver CRISPR sgRNA and perform pooled CRISPR screens in an *Anopheles* cell line. Altogether, we provide a platform for high-throughput genome-scale screening in cell lines from disease vector species.

[1] Department of Genetics, Blavatnik Institute, Harvard Medical School, Boston, MA 02115, USA. [2] Department of Microbiology, National Emerging Infectious Diseases Laboratories, Boston University School of Medicine, 620 Albany Street, Boston, MA 02118, USA. [3] HHMI, Harvard Medical School, Boston, MA 02115, USA. [4] These authors contributed equally: Raghuvir Viswanatha, Enzo Mameli. ✉email: ram@genetics.med.harvard.edu; perrimon@receptor.med.harvard.edu

Mosquito-borne diseases include a vast repertoire of viral, bacterial and parasitic diseases of medical and veterinary importance, with malaria alone causing nearly half a million human deaths each year[1]. Current efforts to fight malaria and other mosquito-transmitted pathogens such as dengue, Zika, Chikungunya and West Nile viruses rely on control of vector populations, mostly by means of insecticides[2,3]. These measures are hampered by ever-increasing insecticide resistance[4,5]. Alternative strategies under current development[6,7] include those based on the use of endosymbiotic bacteria such as *Wolbachia*[8,9], gene drives to suppress wild mosquito populations[10,11], or introduction of disease-refractory mosquitos[12–14].

A key advantage of the introduction of CRISPR-Cas9 technology was the ability to generate large pools of sgRNAs and simultaneously test their effect in mammalian cells. This approach has transformed several areas of cell biology and revealed the function of previously unannotated genes[15]. CRISPR screening in mammalian cells has already provided key insights into the entry and infection mechanisms of numerous toxins, parasites, bacteria and viruses[16,17], including mosquito-borne viruses[18,19]. However, mosquito-borne viruses interact with a distinct set of host factors in the mammalian and insect host. Moreover, mosquito-borne viruses cause fewer cytopathological effects (CPE) in mosquito cells than in mammalian cells, and tend to develop persistent infections in mosquito cells but not in mammalian cells[20–23]. Thus, to better understand the mosquito host genes involved in pathogen interactions, a method for unbiased genetic screening in mosquito cells is needed.

Roughly 20 mosquito cell lines from *Aedes*, *Culex*, and *Anopheles* genera have been established over the last 50 years[24]. These cells are most widely used to propagate and characterize mosquito-borne viruses, including dengue, yellow fever, La Crosse, Japanese encephalitis virus, West Nile, Rift Valley, o'nyong-nyong, Sindbis, and Zika viruses[24]. Studies using these cell lines have revealed dependencies, such as a need for low pH of endocytic compartments for infection[25] and specific host factors[26]. Immune-competent mosquito cell lines are also useful in dissecting the innate immune response[27] and the unique mosquito cellular anti-viral response, which involves the somatic production of PIWI-interacting small RNAs[22,28,29]. Cell lines also provide a platform to propagate viruses or intracellular pathogens[30], and permit in vitro characterization of mosquito-specific drugs[31], toxins[32], viruses[33], and *Wolbachia*[8,34], supporting the development of biocontrol strategies.

RNAi knockdown and CRISPR-Cas9 experiments have been performed in mosquito cells[26,27,35–37] but neither has been applied at genome-scale. Furthermore, previous experiments relied on *Drosophila*-optimized CRISPR reagents[36,37] and only recently, efforts have begun to optimize reagents for mosquitos[35,38,39]. Here, we have developed methods and resources for genome-scale CRISPR screening in mosquito cells. First, we created a bioinformatics portal that allows us to construct targeted mosquito sgRNA libraries. Next, we identified optimal *pol III* promoters for mosquito cell lines. Then, based on our previous work in *Drosophila* cells[40], we used recombination-mediated cassette exchange (RMCE) to deliver complex CRISPR sgRNA libraries to mosquito cells. We demonstrate the robustness of the approach by performing a large-scale pooled CRISPR screen in Sua-5B, a cell line derived from the major African malaria vector *Anopheles coluzzii*[41]. Altogether, we demonstrate that unbiased loss-of-function screens can be performed in mosquito cell lines, setting the stage for genome-scale screens and other studies based on these approaches and resources.

## Results

### A unified resource for ortholog search and batch CRISPR guide design in mosquito species.

To facilitate CRISPR-based genome engineering in mosquitos and provide a batch-mode design resource for pooled CRISPR knockout (KO) screening targeting protein-coding genes, we developed CRISPR GuideXpress (https://www.flyrnai.org/tools/fly2mosquito/web/), an online resource with a number of features. First, CRISPR GuideXpress allows users to input genes from *Drosophila*, which as a model organism for dipterans has a very well-annotated genome, and retrieve the closest mosquito orthologs. Orthology is mapped using an approach similar to DIOPT[42]. Second, CRISPR GuideXpress allows users to retrieve large numbers of sgRNAs targeting multiple genes, as is required for CRIPSR screen library design. Third, CRISPR GuideXpress supports several mosquito species—*An. gambiae*, *An. coluzzii*, *An. stephensi*, *C. quinquefasciatus*, *Ae. aegypti*, and *Ae. Albopictus*—and can be further updated to include additional species (Fig. 1a). Finally, the sgRNA designs are accompanied by pre-computed sets of parameters that are displayed alongside sgRNA sequences and the total search output can be downloaded in table format. Efficiency predictions are calculated based on the 'Housden score'[43] and a machine learning-based analysis of *Drosophila* CRISPR cell screen data[40]. CRISPR GuideXpress also provides a cross-species reference when the same guide targets a homologous gene in one of the other supported species allowing, in some cases, inter-species targeting with the same reagents. For each mosquito species, the sgRNA designs cover ~92-99% of protein-coding genes, and at least ~62-93% of protein-coding genes are targeted by 6 or more high quality sgRNAs (i.e., designs with no predicted off-targets). The number of designs and relative coverage per gene for mosquito genomes is similar to the library used for CRISPR KO screening in *Drosophila* cells[40,44] (Fig. 1b,c). Furthermore, for *An. gambie* and *An. coluzzii*, we incorporated a variant database based on full genome sequences of hundreds of field samples from the *Anopheles* 1000 Genomes Project[45,46] in order to allow for the selection of designs that would avoid common SNPs in wild populations (Fig. 1d).

### Engineering RMCE acceptor mosquito cell lines.

To generate RMCE acceptor cell lines as a platform for CRISPR screens, we first chose well-characterized cell lines from three mosquito species that are susceptible to infection with biomedically important viruses or parasites, and for which genomic, transcriptomic, and small RNA sequencing data exist[29]: Sua-5B from *Anopheles coluzzii*[41] (formerly *An. gambiae* M form[47]), NAMRU2-CQ-01 (also known as Hsu) from *Culex quinquefasciatus*[48], and C6/36 from *Aedes albopictus*[49]. Our previous work in a *Drosophila* cell line showed that CRISPR screens can be conducted by first introducing constitutive Cas9 expression and then transfecting cells with donor sgRNA expression vectors that can integrate into the RMCE loci[40,50]. This way, each cell stably integrates a small number of different sgRNA expression cassettes according to the number of RMCE insertion sites (Fig. 2a). To validate this approach in mosquito cells, we first constructed a series of RMCE lines using a MiMIC vector mobilized from a plasmid in the host genome at low frequency. Modified cells are identified by the presence of an mCherry exon that becomes incorporated into a native gene. mCherry-expressing cells were isolated using fluorescence-activated cell sorting (FACS) and we selected a single, strong mCherry-positive derivative cell-line from each parental line: Sua-5B-IE8 (*Anopheles*), NAMRU2-CQ-01-1.7 (*Culex*), and C6/36-HE8 (*Aedes*). As expected[50], we observe different mCherry distributions in each clonal isolate (Supplementary Fig. 1b–d). In Sua-5B-IE8, mCherry antibodies detected a strong band at

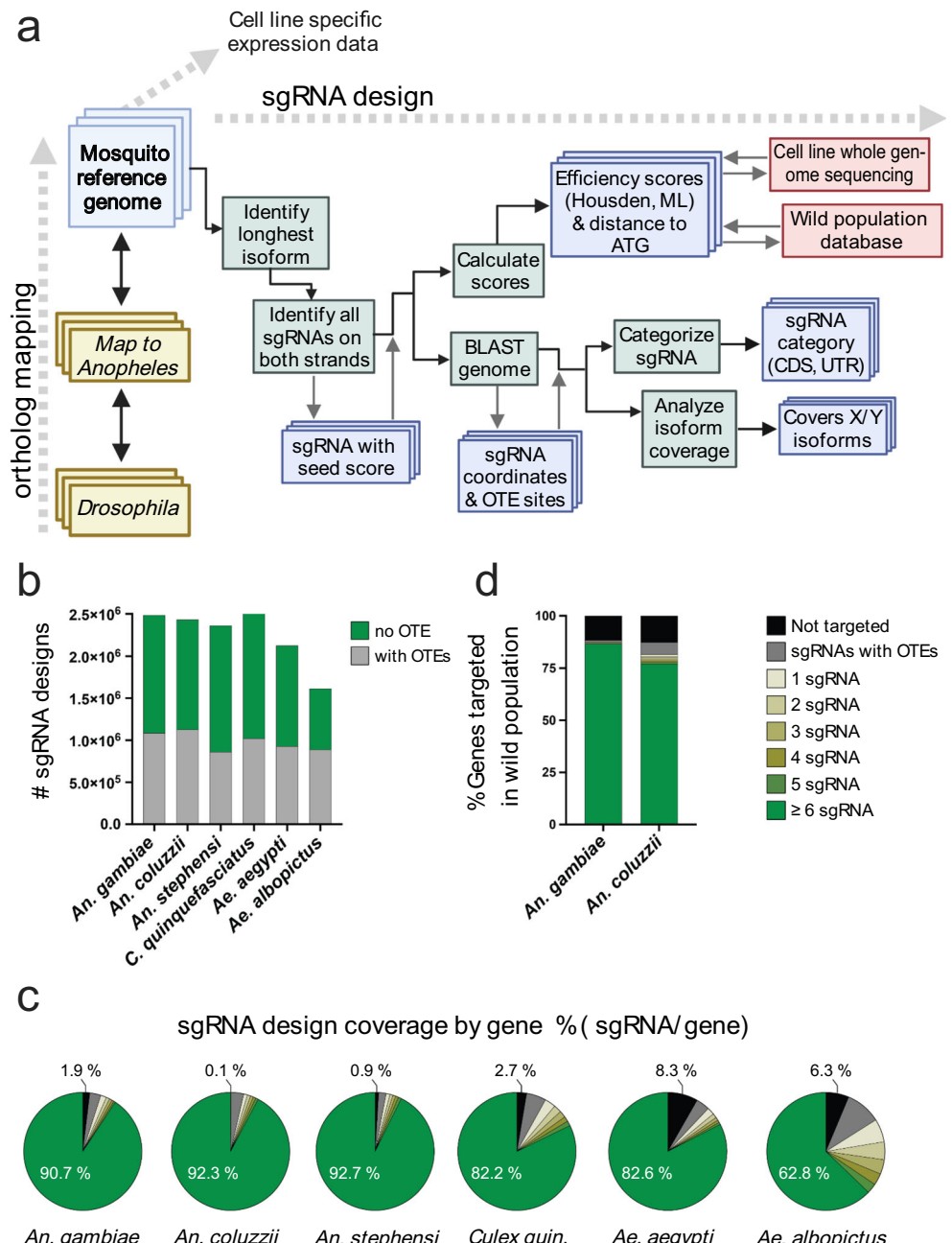

**Fig. 1 CRISPR GuideXpress: an online bioinformatics framework for CRISPR sgRNA design and analysis. a** Features and sgRNA design workflow. Ortholog mapping, cell line-specific expression data, and sgRNA design for six supported mosquito species are integrated at one interface. Genes can be searched individually or in batch mode. Direct ortholog searching is available between *An. gambiae* and other mosquito species or *Drosophila*. After a gene name or ID is entered, the tool retrieves corresponding transcripts and displays precomputed sgRNAs and associated scores. The sgRNAs are computed as follows. The longest isoforms are identified from transcripts. Next, all possible PAMs and associated sgRNA designs on both strands are selected. Each design is then assigned a seed score based on uniqueness of the 12-15 nt 3' sequence (excluding the PAM). For each guide, a BLAST search is used to define specificity (off-target score). Each guide is mapped to the genome and categorized based on the gene region targeted and the respective isoform coverage. All sgRNA designs are evaluated to yield multiple efficiency parameters: 'Housden' score, machine learning (ML) score, and distance from ATG. Additionally, sgRNA designs for *An. gambiae* and *An. coluzzii* are assigned a 'wild population' efficiency score calculated from the Ag1000 Genome project dataset (see methods). To optimize for use in *An. coluzzii* Sua-5B cells, the tool indicates if the sgRNA sequences fully match the Sua-5B whole-genome sequence. (**b**–**d**) Analysis of genome-wide CRISPR KO sgRNA designs targeting protein-coding genes in supported mosquito species. **b** Histogram representing total number of sgRNA designs in two categories: (green squares) "no OTE" (off-target effect), with minimal off-target effects, or (gray squares) "with OTE" within the criteria (see Methods). **c** Genome-wide sgRNA design coverage, showing the percentage of genes targetable by sgRNAs with minimal OTE (light yellow to green), targetable only by sgRNAs with potential OTE (gray), or untargetable (black). **d** Genome-wide sgRNA design coverage by gene (%) in wild populations sampled in the Ag1000 Genome project. % of genes targeted and ranking based on # of sgRNAs/gene, as specified above. For this analysis were considered only sgRNA designs matching ≥ 95% of the wild genome sequences sampled. Source data are provided as a Source Data file. Raw statistics can be found in Supplementary Data 4.

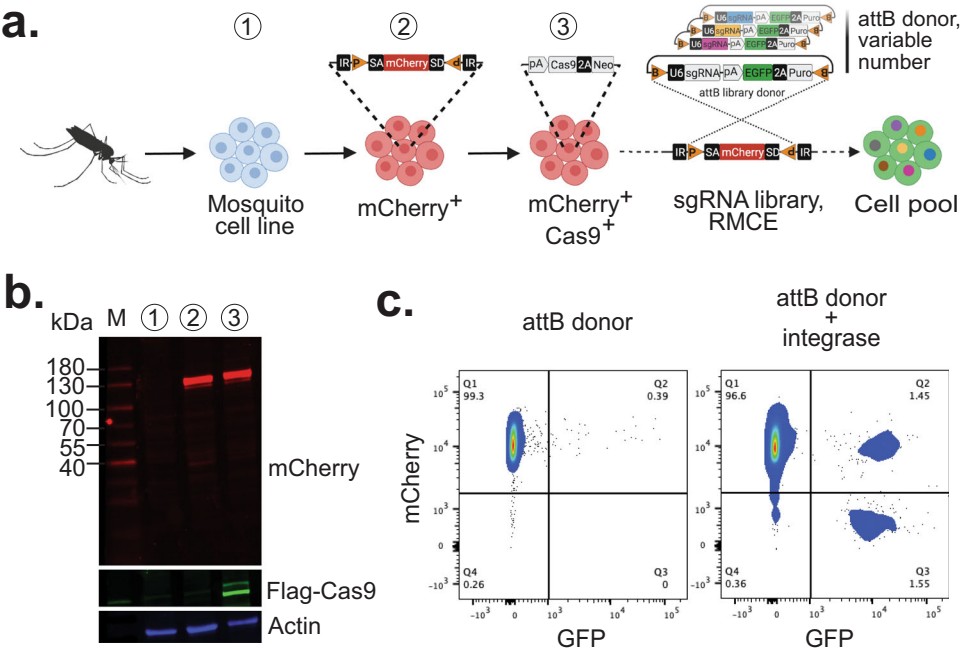

**Fig. 2 Engineering mosquito RMCE cell lines. a** Schematic for building CRISPR screen-ready cell lines. MiMIC technology converts an established mosquito cell line into RMCE acceptor cell line. **b** Western blot of Anopheles Sua-5B cell line along with the steps of engineering into "CRISPR-ready" cell line expressing mCherry and Cas9 proteins. **c** Flow cytometry plot of cells transfected with GFP-expressing attB donor plasmid without or with integrase. ~3% of cells are integrated cells when integrase is provided, compared with 0.39% randomly integrated. Experiments were repeated three times with similar results. Uncropped images of the gel are provided as a Source Data file.

~120 kDa, indicating that a protein fusion had been created using the approach (Fig. 2b). RMCE acceptor sites allow recombination via the bacteriophage ΦC31 integrase[50,51]. As a second step, we stably transfected pAct::Cas9-2A-Neo (driven by the *Drosophila* actin promoter) and characterized the resulting cell line, namely Sua-5B-IE8-Act::Cas9-2A-Neo. For validation of RMCE, we first performed next-generation sequencing of the cell line and mapped all MiMIC transposon insertions, revealing 5 insertion sites (Supplementary Fig. 1e). Following recombination, 1-5 RMCE events are expected per cell. Pooled screens can successfully reveal hits with guide multiplicities as high as 10, and strategies exist to eliminate noise in high multiplicity screens[52–54]. We measured the recombination frequency by transfecting either an attB donor alone or along with an integrase expression vector (Fig. 2c). Following a month of passaging each cell population (without any selection agent), the proportion of mCherry positive cells was ~3%, compared with 0.39% without integrase. Thus, as roughly 85% of cells underwent site-specific recombination, Sua-5B-IE8 cells are suitable for RMCE.

**Identifying optimal U6 promoters for CRISPR KO in mosquito cells**. An incompletely addressed challenge for CRISPR genome engineering in mosquitos is the identification of optimal *pol III* promoters for heterologous expression of sgRNAs[35,38,39,55,56]. We performed a side-by-side evaluation of eleven *pol III* promoters from four mosquito species, as well as a consensus sequence, in each of the three mosquito cell lines. To choose promoters, we first used BLAST and multiple sequence alignments to identify orthologs of the *Drosophila* U6-2 (*snRNA:U6:96Ab*) promoter and chose eleven orthologous promoters from U6 snRNAs of *Anopheles*, *Culex*, or *Aedes* (Fig. 3a). When possible, we selected a minimum of three promoters per species, prioritizing U6 promoters that contain an intact *pol III*

bipartite promoter motif and for which RNA-seq data suggests they are expressed in cell lines and in adult tissues (see Methods). These were synthesized and inserted into pLib6.4[40,44] to generate a suite of vectors for the expression of sgRNA under the control of different U6 promoters (Supplementary Data 1).

Mosquito cells with genomically-encoded mCherry allowed us to use a flow cytometry-based dual reporter assay to directly compare KO efficiency in cells expressing the same sgRNA from different U6 promoters (Fig. 3b, Supplementary Fig. 2, Supplementary Data 1). In this strategy, we test U6 promoter strength by measuring the ability of the downstream sgRNA to suppress mCherry. Specifically, we co-transfected mCherry expressing cells with a Cas9 expression vector and a plasmid containing a mCherry-targeting sgRNA driven by a variable U6 promoter. The U6 promoter plasmid co-expressed GFP as an indicator of transfection. After gating cells with GFP expression, the ratio of mCherry⁻ cells is used to determine KO efficiency. This gating strategy allows us to mitigate differences in transfection efficiency. An improvement of this approach over a plasmid-based dual reporter assay is that mCherry is genomically encoded rather than an episomal target, revealing repair outcomes that would be expected at a native gene. Although the number of insertions and expression level of mCherry vary between cell lines, this approach permits the comparison of different U6 promoters within the same cell line. In *Anopheles* cells, all mosquito promoters tested elicited measurable KO, whereas *Drosophila* promoters failed (Fig. 3c, Supplementary Fig. 2b). The native promoters (AGAP013695, AGAP013557) along with *Culex* CPIJ039596 and *Ae. aegypti* AAEL017774 showed the strongest activity, achieving approximately 75% KO efficiency relative to controls. In particular, AAEL017774 (mean = 81.3 SD ± 1.9) and AGAP013695 (mean = 76.6 SD ± 3) were the most efficient. The remaining promoters have moderate to low activity, and the mosquito consensus promoter performed similarly to the native promoters. In the *Culex* cell line, we observed a more uniform activity of mosquito U6

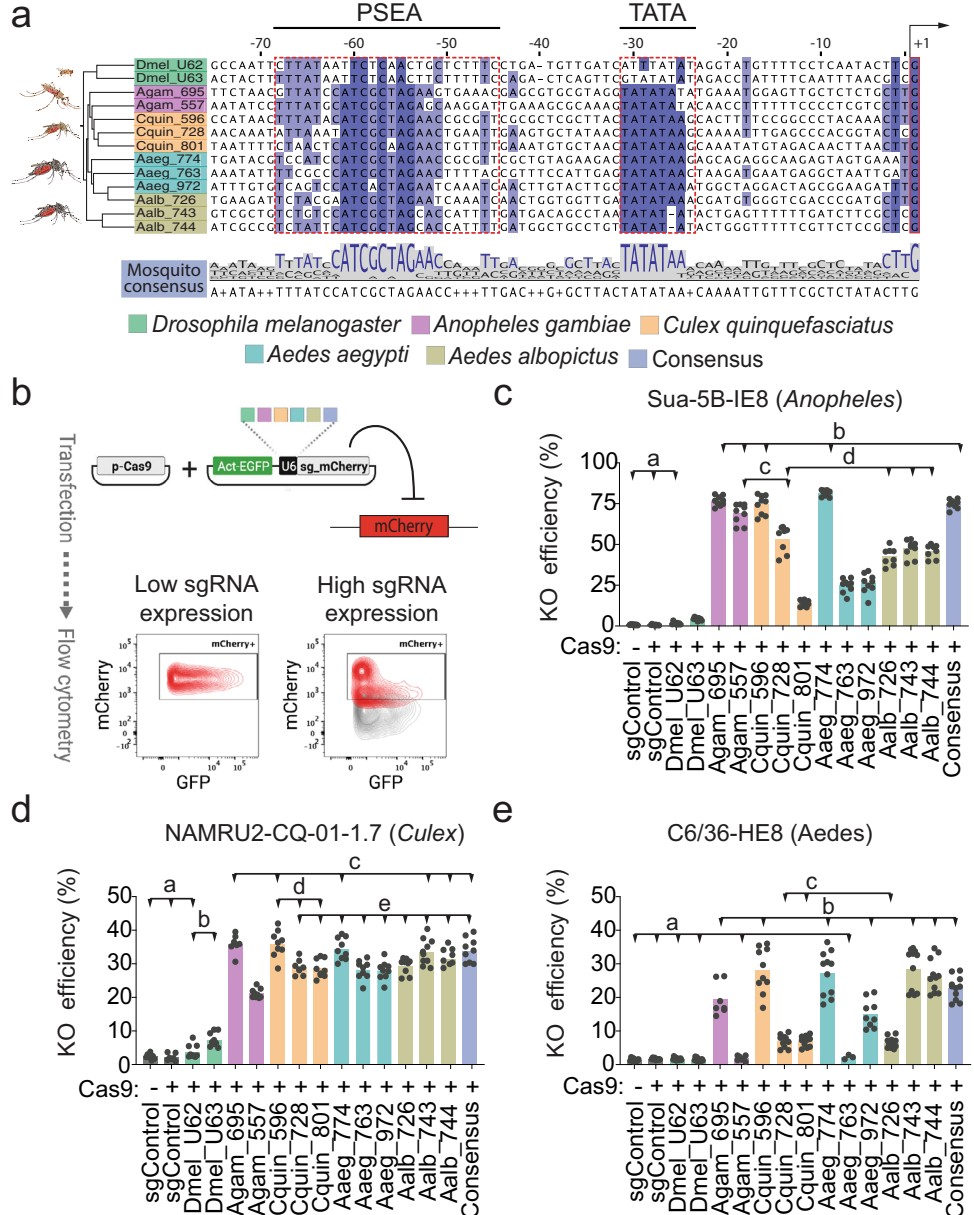

**Fig. 3 Identification of U6 promoters for sgRNA expression in mosquito cells and evaluation of CRISPR KO efficiency. a** Multiple alignments of selected U6 promoters highlighting the metazoan *pol III* promoters bipartite structure and a mosquito consensus sequence. The first transcribed base of the U6 snRNA, the TATA box, and proximal sequence element A (PSEA) are boxed in red. The consensus sequence derived from mosquito sequences was used to design a synthetic U6 promoter. **b** Flow cytometry-based assay for evaluation of CRISPR KO efficiency with different U6 promoters. Engineered mosquito cells expressing mCherry were co-transfected with a plasmid expressing Cas9 and a GFP reporter plasmid expressing the sgRNA targeting mCherry under the control of different U6 promoters. After transfection, cells were passaged for up to 12 days and then analyzed by flow cytometry. Only GFP+ cells were considered in the analysis. Relative KO efficiency was obtained by calculating the ratio of mCherry- cells over total GFP transfected cells. **c–e** Evaluation of U6 promoter-specific CRISPR-KO efficiencies in three mosquito cell lines. Histogram bars represent the mean, dots represent the distribution of multiple reps obtained from 3 independent experiments. Histogram colors denote the species of origin of the U6 promoters analyzed, shown with abbreviation of species name and three last letters of the corresponding Vectorbase gene ID. sgControl = pLib6.4-Agam_695 U6 expressing the empty BbsI cassette was used as control. Statistical analysis was performed using Brown-Forsythe and Welch ANOVA tests followed by Dunnett's multiple comparison test. Lowercase letter groupings denote differences not significant ($P_{Dunnett} > 0.05$). All differences between samples of different groupings are significant ($P_{Dunnett} < 0.05$). Raw data, detailed descriptive statistics, and statistical analysis, including sample number and calculated P values for each comparison, are reported in Supplementary Data 2 and Source Data files.

promoters, with an overall mean KO efficiency of about 30% (Fig. 3d, Supplementary Fig. 2b). Notably, the results for CPIJ039596 obtained using this assay were slightly lower but overall comparable to CRISPR allele editing efficiency as verified by deep sequencing for the same promoter in our previous study[38].

The most effective U6 promoters in *Ae. albopictus* C6/36-HE8 cells were the native promoters AALF029743-4 (mean = 28.6 SD ± 6.1; mean = 26.4 SD ± 4.9), *Culex* CPIJ039596 and *Ae. aegypti* AAEL017774, with about 27% mean KO efficiency (Fig. 3e, Supplementary Fig. 2b). Interestingly, *Culex* CPIJ039596,

*Ae. aegypti* AAEL017774, and the consensus mosquito promoters performed consistently within each species, suggesting that these promoters might work in other mosquito species for which CRISPR reagents have not yet been optimized. Of the two *Drosophila* U6 promoters tested, only U6:3 resulted in significant KO effects in *Anopheles* and *Culex* cell lines but with very low efficiency (mean = 4.3 SD ± 0.9; mean = 7.4 SD ± 2). A secondary analysis of the flow cytometry data, performed using the variation of the median fluorescence intensity (MFI) of the mCherry signal within the GFP$^+$ cells, confirmed the relative changes in KO efficiencies (Supplementary Fig. 2). These results are in accordance with the overall evolutionary distance between the species and U6 promoter sequence average distance and corroborate previous in vitro[35,38,39,55] and in vivo[38,56] results. Surprisingly, mosquito U6 promoters displayed more uniformly high activity in *Drosophila* cells. In these cells native *Drosophila* promoters achieved 90-95% (U6-2 mean = 92.6 SD ± 0.7; U6-3 mean = 94.4 SD ± 0.5) KO efficiency, while mosquito promoters performance ranged from a minimum of ~40% (CPIJ039801 mean = 41.24 SD ± 1.6) to a maximum of ~70% (AALF029743 mean = 68.15 SD ± 1.3) KO efficiency (Supplementary Fig. 2c).

**CRISPR KO produces an observable phenotype in Anopheles cells.** We next asked whether our *Anopheles* CRISPR screening platform results in penetrant, visible phenotypes. In Sua-5B-IE8-Act::Cas9-2A-Neo cells, we asked whether a visible phenotype can result from introducing a sgRNA expression cassette targeting *Rho1* (AGAP005160), which is necessary for the completion of cytokinesis, driven by the optimal U6 promoter Agam_695 (Fig. 3c). Previous reports have shown that knockdown of *Rho1* by RNAi in *Drosophila*[57] or *Anopheles*[27] cells results in a modest size increase (~2-fold) due to cell growth without division, and *Drosophila* cells expressing CRISPR sgRNAs targeting *Rho1* become dramatically enlarged due to complete loss of *Rho1*[40]. To test the novel *Anopheles* cell-based CRISPR system, we transfected sgRNAs targeting the *Anopheles Rho1* ortholog AGAP005160 and observed transfected cells after several days of selection. We found that *Rho1* sgRNA-expressing cells, but not control cells, became enlarged up to 6-fold (Fig. 4a, b). We used T7 Endonuclease I assays to confirm editing of the *Rho1* locus (Fig. 4c). Enriching for the sgRNA-expressing cells resulted in greater editing, as would be expected if the editing frequency was limited by a low percentage of sgRNA transfection (Fig. 4c). These results clearly demonstrate that the Sua-5B-IE8-Act::Cas9-2A-Neo 'CRISPR-ready' cell line can yield highly penetrant phenotypes, suggesting that the system is compatible with CRISPR screening.

**Validation of the CRISPR screening platform in Anopheles cells.** To directly test applications of the CRISPR screening platform at a large scale in mosquito cells, we first chose five genes that had previously been shown to be drug-resistance factors in *Drosophila* cells[40] and used CRISPR GuideXpress to design a library targeting their orthologs in *Anopheles coluzzi*. Target genes included *Anopheles* orthologs of *FKBP12* (AGAP012184), which encodes the cellular binding partner of the mTOR inhibitor rapamycin; *EcR* (AGAP028634) and *usp* (AGAP002095), which encode mediators of an antiproliferative transcriptional response to treatment with ecdysone; and *PTP-ER* (AGAP007118), which encodes a negative regulator of the mitogen-activated protein kinase (MAPK) signaling cascade that can be suppressed by treatment with the MEK inhibitor trametinib (Fig. 5a; Supplementary Data 3). In total, 3,487 sgRNAs were synthesized and cloned into pLib6.4-Agam_695 containing the strong *Anopheles* U6 promoter and transfected into *An. coluzzii* Sua-5B-IE8-Act::Cas9-2A-Neo cells in the presence of ΦC31 integrase to facilitate recombination, then selected for 16 days in puromycin-containing media with continuous passaging every four days. A theoretical copy number of 1000 cells per sgRNA was maintained during all passages. For the selection screens, the cells were grown for an additional 30 days in the presence of rapamycin, ecdysone (20-hydroxyecdysone), or trametinib (Fig. 5b). Then, genomic DNA was collected, and the sgRNA-containing locus was PCR amplified, barcoded, and analyzed by next-generation sequencing (NGS). Guides targeting *FKBP12* (AGAP012184) were clearly enriched by treatment with rapamycin but not in untreated, ecdysone, or trametinib growth conditions (Fig. 5c). Sequence analysis of the *FKBP12* locus in the Sua-5B-IE8-Act::Cas9-2A-Neo cell line revealed a coding variant in the cells relative to the reference genome (AgamP4) that results in single-base mismatches between a subset of three sgRNAs designed to target the *FKBP12* locus. Unlike no-mismatch guides, these mismatched guides were not selected in rapamycin treatment conditions (Fig. 5d). Similar observations were made for the set of guides targeting *usp* (Supplementary Fig. 3; Supplemenary Data 3). After observing these single nucleotide polymorphisms in specific genes, we referred to the whole-genome-sequence of the Sua-5B-IE8-Act::Cas9-2A-Neo 'CRISPR-ready' cell line and added a variant analysis to CRISPR GuideXpress, giving users the option to exclude these variants from sgRNA designs (Fig. 1a). Importantly, for all three screens, we found significant and selective enrichment for the orthologs of the expected genes: *FKBP12* ($p < 4.99E-06$ in rep # 1, $p < 4.99E-06$ in rep #2) for rapamycin, *EcR* ($p < 1.50E-05$ in rep # 1; $p < 3.39E-06$ in rep # 2) and *usp* ($p < 4.99E-06$ in rep # 1; $p < 4.32E-17$ in rep # 2) for ecdysone, and *PTP-ER* ($7.32E-17$ in rep # 1; $p < 1.14E-09$ in rep # 2) in trametinib (Fig. 5e; Supplementary Fig. 3; Supplemenary Data 3). This was driven by consistent enrichment of

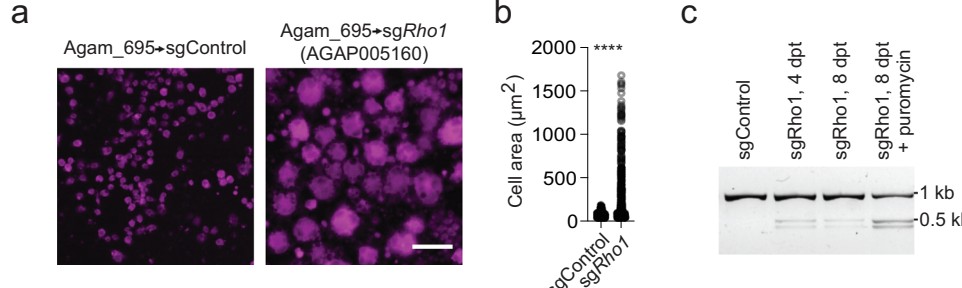

**Fig. 4 CRISPR produces highly penetrant phenotypes in the *Anopheles* CRISPR-ready cell line. a** Sua-5B-IE8-Act::Cas9-2A-Neo cells were stably transfected with pLib6.4-Agam_695 donor vector encoding a *Rho1* sgRNA, leading to a highly penetrant failure in cytokinesis and dramatically enlarged cell area **b** Quantification of cell enlargement (****$P < 0.0001$, unpaired *t*-test, two-tailed, $t = 13.45$, df = 890, sgControl $n = 509$, sgRho1 $n = 383$). mCherry signal (purple), scale bar, 50 μm. **c** T7 Endonuclease I assay confirming genomic editing in cell populations following 4 or 8 days post-transfection (dpt) without or with puromycin selection to enrich for cells expressing *Rho1* sgRNA. Experiments in (**a**, **b**) and (**c**) were repeated three or more times with similar results. Source data for **b** is provided in the Source Data file.

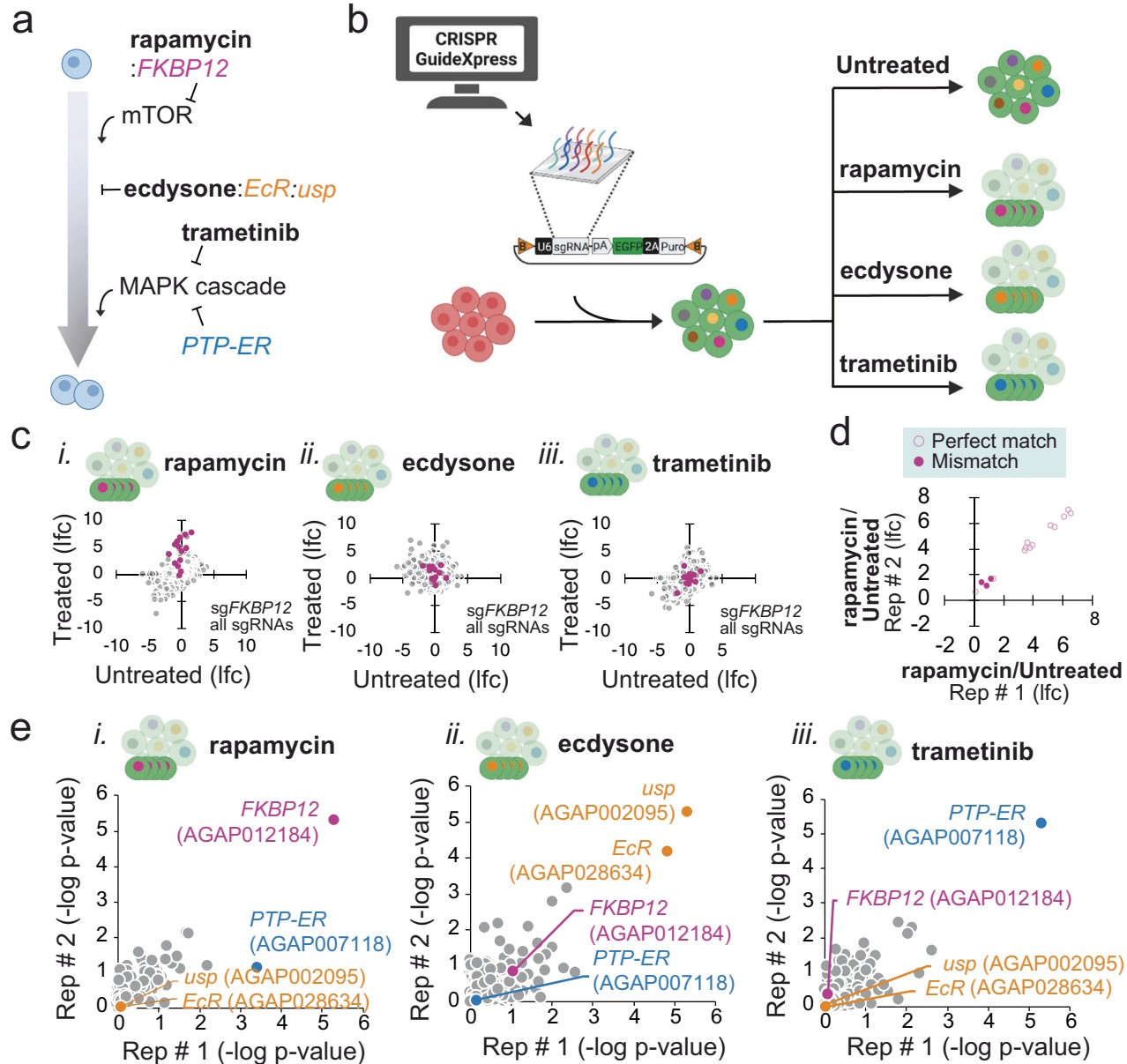

**Fig. 5 Pilot pooled CRISPR drug resistance screens in *Anopheles* cells. a** Schematic of proliferation-related pathways used to validate the screening approach. Rapamycin binding to FKBP12 inhibits mTOR, which is necessary for proliferation. Ecdysone binding to EcR and Usp inhibits proliferation. Trametinib inhibits the mitogen-activated protein kinase (MAPK) signaling cascade, which normally promotes proliferation. PTP-ER is a potent endogenous negative regulator of MAPK. **b** We used CRISPR GuideXpress to design a library of 3487 sgRNAs against mosquito orthologs of *FKBP12, EcR, usp*, and *PTP-ER*. These were cloned into a library donor vector (pLib6.4-Agam-695) and integrated into Sua-5B-IE8-Act::Cas9-2A-Neo cells. The cells were left untreated or treated with the indicated drugs. **c** Endpoint sgRNA readcounts compared to plasmid readcounts show that more than half of all *FKBP12* sgRNAs were enriched following growth in rapamycin but not ecdysone or trametinib. **d** FKBP12 sgRNAs predicted to target the reference genome, AgamP4, but with mismatches as compared to the Sua-5B-IE8-Act::Cas9-2A-Neo genome failed to enrich following rapamycin treatment. **e** Robust rank aggregation (RRA) analysis of sgRNA readcount data from all screens shows that *FKBP12* was selectively enriched after rapamycin treatment, *EcR* and *usp* were selectively enriched after ecdysone treatment, and *PTP-ER* was selectively enriched in trametinib treatment (two biological replicates). Source data are provided as a Source Data file. Raw data can be found in Supplementary Data 3.

multiple sgRNAs targeting each selected gene in each drug treatment regime: under rapamycin selection, *FKBP12* sgRNAs made up 9 of the top 20 enriched sgRNAs; under ecdysone selection, *EcR* made up 5 and *usp* made up 5 of the top 20 sgRNAs in the screen; under trametinib selection, PTP-ER made up 9 of the top 20 enriched sgRNAs (Supplemenary Data 3). We note, however, that we did not

observe enrichment for a candidate *EcI* ortholog, AGAP006638 ($p < 0.78292$ in rep # 1; $p < 0.80726$ in rep. # 2) (Supplementary Fig. 3; Supplemenary Data 3). These results suggest that using the RMCE approach, optimized U6 expression, and CRISPR sgRNA design pipeline we have developed will make it possible to efficiently conduct massively parallel genetic screens in mosquito cells.

## Discussion

Here we have developed tools for performing large-scale pooled CRISPR KO screens in mosquito cell lines and carried out a large-scale genetic screen in *Anopheles* cells. To establish the platform, we created a bioinformatic tool for batch sgRNA design; experimentally tested U6 promoters to identify those with high activity; cloned several CRISPR plasmid vectors and a large-scale sgRNA library targeting mosquito genes; modified mosquito cell lines for RMCE; and demonstrated, as expected from previous studies, that *Rho1* KO in mosquito cell lines causes a strong, visible phenotype useful for assessing the efficiency of CRISPR modification. In a large-scale pooled CRISPR KO screen in *Anopheles* cells, we were successful in specifically enriching for genes that when knocked out were expected to provide resistance to one of three different experimental treatment conditions. Notably, the screen results validate the function of the predicted *Anopheles* orthologs of the *Drosophila FKBP12*, *EcR*, *usp*, and *PTP-ER* genes.

The online portal we introduce here, CRISPR GuideXpress, can be used to design single guides, genome-wide libraries, or focused libraries of variable size. As a tool for designing sgRNAs for individual CRISPR KOs, GuideXpress enables prioritization of guide designs by several parameters, including mismatches relative cell lines or wild mosquito genomes. As a tool for batch sgRNA design, focused libraries, such as the one we created (Fig. 5), could have several immediate applications in mosquito research. Several studies have generated genesets from proteomic or differential expression analyses (e.g., host proteins that interact with viral proteins[58], or genes up- or down-regulated in response to pathogen infection[26,34,37,59,60]). Focused CRISPR screening based on these genesets can provide functional validation of these data. The user can also provide high numbers of sgRNAs per gene, reducing noise particularly for challenging screens.

In addition, this study also addresses a broader lack of CRISPR tools for mosquito research. Basing CRISPR KO constructs on reagents optimized for use in *Drosophila* has worked for generating CRISPR KO mosquito cells[36,37]. However, our results testing U6 promoters in this and previous work including our group[38,39,55] suggest that species-specific optimization is worthwhile. Studies in mammalian cells have shown that empirically optimized sgRNAs lead to reduced off-targets and increased efficiency[61], and our studies enable the application of this strategy to mosquito genomes. Finally, our studies provide at least two high-expression U6 promoters for each mosquito species. Additional mosquito U6 promoters with lower activity could still be useful in applications where sgRNA dosage needs to be controlled. Furthermore, having multiple U6 promoters enables the combinatorial expression of sgRNAs in the same cell, reducing the chances of recombination between identical U6 promoter sequences[62,63]. Interestingly, to our knowledge, we report the first comparison of mosquito U6 promoters in *Drosophila* cells, identifying at least three promoters with significant activity, expanding the array of tools for multiplexed CRISPR targeting in flies[64] (Supplementary Fig. 2c). Finally, our work provides a platform that could accommodate the creation of single KO cell lines or KO pools of variable size, complementing and expanding substantially the tools currently available[35] for KO studies in mosquito cell lines.

Further development of the screening strategy is likely to improve the platform in the future. First, although ΦC31 RCME efficiency is high in *Anopheles* cells (~85%), the initial transfection efficiency is low compared with *Drosophila* cells. As a result, a larger number of cells must be transfected, using a larger amount of a costly transfection reagent to achieve a comparable screen. Optimizing transfection efficiency, such as by using electroporation, has the potential to reduce screening costs without changing screen outcomes. Second, the finding that there

are 1 to 5 RMCE cassettes per cell following transfection of the pooled library raises the possibility of "passenger effects" during selection that could reduce the resolution of the screen. Even in conditions of high multiplicity delivery (up to 10 sgRNAs per cell), high-quality screens can still be conducted by applying recently developed strategies[52–54]; mitigating these effects could be important, as this could reduce the false-discovery rate (FDR) and increase reproducibility. Third, although our screen results verify that our approaches to identification of mosquito orthologs of *Drosophila* genes and to sgRNA design are valid, there is room for improvement in sgRNA design. For example, we would like to incorporate cell line genome data for additional cell lines. Moreover, we and others have learned that large-scale screen data provide lists of 'good' and 'bad' sgRNA designs for genes that were positive in the screen data and as such, can be used to derive sgRNA design rules. Thus, as we accumulate more large-scale screen data, we expect to iteratively improve our CRISPR GuideXpress resource. Finally, the approach should be extensible to screens based on other CRISPR systems, including CRISPR activation and CRISPR interference.

Importantly, the new ability to perform pooled CRISPR KO screens in mosquito cells will facilitate screens for essential genes and for genes that confer sensitivity or resistance to any treatment that slows growth or results in cell death, including insecticides, biological toxins, and other agents, further contributing to functional annotation of mosquito genes. A screen for resistance to ecdysone-induced cell death in *Drosophila* cells, for example, revealed novel ecdysone pathway components, including a previously uncharacterized transporter for ecdysone[65]. The information gained from these screens can also inform our ability to control mosquito populations or infection of mosquitos with human pathogens. For example, the screening platform can be used to identify conserved or species-specific essential genes, which in turn could be used in the design of gene drives[66] and/or for the development of new and potentially highly targeted insecticides. In addition, screens in mosquito cells have the potential to increase our knowledge of host-pathogen interactions. Genome-scale CRISPR and RNAi screens have been used in the past to investigate interactions between mosquito-borne viruses and mammalian[18,19] or *Drosophila* cells[67,68]. Similar work in mosquito cells has been limited to targeting a few genes using RNAi[26] or drug treatments[58]. The ability to perform large-scale CRIPSR KO screens in mosquito cells opens the door to potentially novel findings regarding the interaction of mosquito host cells with viral and other pathogens and holds great potential for aiding the multifront effort to control mosquito-borne diseases.

## Methods

**Identification of mosquito U6 promoters.** U6 snRNAs are conserved eukaryotic non-coding RNAs that take part in the formation of the catalytic core of the spliceosome, while their promoters and associated *pol III* transcriptional machinery show divergence even between closely related species. We used the *Drosophila* U6:2 (CR32867) snRNA sequence as a query to perform a BLAST search of mosquito reference genomes, using the Vectorbase interface (now VEuPathDB at https://vectorbase.org/vectorbase/app). Next, all sequences of the identified orthologs within each species were subjected to multiple alignments (using ClustalΩ at www.ebi.ac.uk[69]) including the full snRNA and 500 bp of upstream sequence, allowing us to visualize and exclude sequences lacking conserved portions of the *pol III* bipartite promoter motif (i.e., the PSEA and TATA). As a secondary criterion for selection, we relied on expression levels reported on RNA-seq data publicly available for each species, including tissue-specific RNA-seq of adult mosquitos or cell lines (available on vectorbase/VEuPathDB, e.g., from the Arthropod Cell Line RNA Seq initiative at the Broad Institute). This data, although not fully reliable in reporting expression levels of small non-coding RNAs, was informative in narrowing our choice to putatively expressed snRNAs. Third, we consulted literature reports of the activity of these promoters. After selecting up to three U6 promoters for each species, we performed a second inter-species alignment and selected for each promoter a region upstream of the TSS of an arbitrary length ranging from 144 to 237 bp. In addition, we selected a 99 bp mosquito consensus sequence derived from the alignment. The length of the AGAP013695 promoter was chosen

based on previous work from Konet et al.[55] in cells and Hammond et al.[10] in adult *Anopheles* mosquitos. The alignment shown in Fig. 3a was obtained by aligning the 250 bp upstream of *Drosophila* U6-2 and U6-3 promoters to the selected mosquito sequences (only a 75 bp region is displayed). The mosquito consensus sequence is shown at the bottom. Jalview Version 2[70] was used to visualize the alignment and to infer the phylogenetic hierarchy based on the average sequence distance of the snRNA and the 75 bp upstream sequence. Additional information about the U6 promoter sequences used for library vector construction is available in Supplementary Data 1.

**Cloning procedures.** In order to build library vectors expressing gRNAs under mosquito U6 promoters we synthesized ~500 bp gBlocks (Integrated DNA Technologies, Inc.) containing in order: the selected U6 promoter sequences; a BbsI cloning site (for gRNA insertion); the gRNA scaffold sequence including an at least 8-T termination sequence[71]; 30 nucleotides of the native termination sequence, and an additional portion of the termination sequence derived from the 3' end of the *Drosophila* snRNA-U6-96AB (CR32867). The gBlocks were sub-cloned into pCR™-Blunt II-TOPO® (Invitrogen) or directly digested with BstBI/KpnI and sub-cloned into pLib6.4 (EGFP reporter) or pLib6.4B (EBFP2 reporter) by replacing the entire gRNA expressing cassette. This resulted in 24 new library vectors harboring the mosquito gRNA expressing cassettes. An additional version of the pLib6.4 attB donor library vector, named pLib6.4B, was obtained by replacing EGFP with EBFP2 using overlap extension PCR with megaprimers as described in Bryksin et al.[72] First, PCR was performed to amplify EBFP2 from pEBFP2-Nuc (Addgene #14893) using primers 14-15. Second, overlap extension PCR was performed using as megaprimers directly the EBFP2 amplicon to mutagenize the EGFP in pLib6.4. A portion of the U6-3 promoter from pCFD3 (PMID: 25002478) was cloned into pLib6.4 to generate pLib6.6. The same strategy, using the same primers, was also used to obtain pLib6.6B from pLib6.6, as well as EBFP2 and EGFP versions of the pSL1180-HR-PUbECFP plasmid (Addgene #47917). In order to generate the EGFP version, megaprimers were obtained by PCR amplification of EGFP from the pLib6.4 vector using the same primers (14-15). Insertion of the gRNA sequence targeting mCherry or Rho GTPase in pLib vectors was performed following an established protocol[73] that allows sgRNA oligos with BbsI sites to be annealed and then ligated into BbsI-digested pLib vectors (primers 1-13). pDmAct5C::Cas9-2A-Neo plasmid was built from Ac5-STABLE1-Neo (Addgene # 32425) by restriction/ligation cloning, replacing GFP with SpCas9, *Drosophila* codon-optimized from pl018[43]. pAaePUb::Cas9-2A-Neo was built in multiple steps. First, the Ae. aegypti polyubiquitin promoter[74] (AAEL003888) was amplified from pSL1180-HR-PUbECFP (Addgene #47917) with primers 16-17 and assembled with EcoRV/SphI linearized pAW (*Drosophila* gateway vector collection) using Gibson assembly (New England Biolabs) resulting in pAePUbW destination vector, where the *Drosophila* Actin5C promoter was replaced with the *Aedes* polyubiquitin promoter. Second, the Gateway cassette in pAePUbW was removed by restriction with EcoRV/PmeI and replaced with Cas9-2A-Neo obtained with double restriction with BsaI/SalI from DmAct5C::Cas9-2A-Neo. When ligation-incompatible restriction extremities where generated, we performed blunting of 3' overhangs using T4 DNA polymerase (New England Biolabs) and proceeded with ligation of blunt ends. Cloning steps involving PCR amplification were performed using Q5 Hot Start High-Fidelity 2X Master Mix (New England Biolabs). All constructs were verified by targeted Sanger sequencing of the modified regions. All primers and plasmids used in this work are listed in Supplementary Data 1.

**Cell lines.** The *Anopheles coluzzii* cell line Sua-5B (RRID:CVCL_RQ24) and *Culex quinquefasciatus* NAMRU2-CQ-01 (RRID:CVCL_1B68, also known as Hsu) cell line, were kind gifts from Flaminia Catteruccia (Harvard T.H. Chan School of Public Health, Boston MA) and Nelson Lau (Boston University, Boston MA), respectively. The *Aedes albopictus* C6/36 (RRID:CVCL_Z230) cell line was from the Colpitts laboratory. The *Drosophila melanogaster* S2R+ -MT::Cas9 (RRID:CVCL_UD30)[40,44] cell line was from the Perrimon laboratory. All cell lines were maintained at 25 °C in Schneider's medium (Gibco), 1x Penicillin-Streptomycin (Gibco), and 10% heat inactivated fetal bovine serum (Gibco). In addition, the media for the *Culex* cell lines was supplemented with 1x MEM NEAA (Gibco). The *Anopheles coluzzii* Sua-5B cell line was authenticated by diagnostic PCR and variant calling analysis of cell-specific whole genome sequence data. Diagnostic PCR was performed following the protocol specified in Santolamazza et al.[75] for *Anopheles* M/S molecular form discrimination (primers 20-21), confirming the presence of the SINE200 insertion specific to the M form (currently recognized as the *Anopheles coluzzii* species). Variant calling analysis (see "Genome variants" below) was performed by comparing whole-genome sequence data from the Sua-5B 'CRISPR-ready' cell line (see "Orthology mapping, sgRNA design, and variant analysis" below) to *Anopheles gambiae* (AgamP4) or *Anopheles coluzzii* (AcolM1.8) genome sequence. We observed significantly more variants when the cell line sequence was compared with the *Anopheles gambiae* genome than when it was compared to *Anopheles coluzzii* (Agam variants = 4,983,818; Acol variants = 4,439,680; Δ = 544,138).

**Cell line engineering.** Native cell lines were first transformed by transposition of an attP-flanked mCherry reporter cassette following an established protocol

(Fig. 2a; Supplementary Fig. 1)[50]. The reporter cassette containing both splice donor and splice acceptor sites, flanked by two inverted ΦC31 attP sites that allow for subsequent RMCE in the presence of the ΦC31-integrase and an attB-flanked donor cassette. Successful insertion through Minos transposition is revealed by mCherry fluorescence, as mCherry is spliced into the trapped gene as an artificial exon. This system offers the advantage to select for insertions in transcriptionally active regions of the genome and acts as a docking pad for the delivery of complex sgRNA libraries through ΦC31 RMCE between attP sites in the cassette (genome) and attB sites of the library donor plasmid. Moreover, this system allows for monitoring of the rate of recombination by flow cytometry or microscopy as cells integrating the library cassette will lose mCherry fluorescence and report fluorescence from the exchanged library vector. Briefly, cells were co-transfected with Actin5C-Minos transposase vector and three separate MiMIC mCherry plasmids (encoding mCherry in each reading frame) in a single transfection mix (molar ratio 1:0.3:0.3:0.3) using Effectene (Qiagen) and following the manufacturer's instructions for 12-well format. After transfection, cells were passaged two times, expanded, and single-cell sorted into 96-well plates by fluorescence-activated cell sorting (FACS) to obtain clonal mCherry-trapped populations. FACS was performed using a FACSAria or MoFlo Astrios (BD) with a 100 um nozzle at 20 psi (Harvard Medical School, Immunology Flow Core). For each cell line, multiple clonal populations were expanded and assessed based on signal intensity and subcellular localization of the mCherry reporter (example in Supplementary Fig. 1). Selected Minos-transformed clones yielded the mCherry positive mosquito-derived cell lines we have named Sua-5B-IE8 (CVCL_B3N2), NAMRU2-CQ-01-1.7 (CVCL_B3N4) (or Hsu-1.7), and C6/36-HE8 (CVCL_B3N5), which were used for CRISPR KO efficiency dual-reporter assays of U6 promoter activity or further modified by stable transfection with Cas9 expressing plasmid. In particular, the *Anopheles* Sua-5B-IE8 clone was transfected with pAct::Cas9-2A-Neo and selected in 400 ug/ml Geneticin (Gibco) for 30 days to obtain our *Anopheles* CRISPR-ready cell line, Sua-5B-IE8-Act::Cas9-2A-Neo (CVCL_B3N3).

**Western blotting.** Semiconfluent cells were washed twice in cold PBS and lysed directly in flasks using RIPA lysis buffer (Pierce #89900) supplemented with protease inhibitor cocktail (Millipore Sigma #S8830) and incubated on ice for 20 m. Whole-cell lysate was first centrifuged at 20 K x *g* at 4 C for 10 m. The supernatant was collected and passed through a 0.45 μm filter column (Corning, #CLS8162). BCA assay (Pierce, #23225) was used to determine protein amounts. Protein extract was added of sample buffer (Bio-Rad, #1610747) and boiled 5 m at 95 C. 20 μg/lane of protein preparation were run on a reducing Tris-Glycine PAGE 4–20% (Bio-Rad, #4561093) and blotted onto 0.45 μm nitrocellulose membrane (Bio-Rad, #1620094). Membrane was blocked for 30 m in blocking buffer (Pierce, #37572) and incubated overnight at 4 C in PBS containing 0.1% Tween 20 (PBS-T) and 10% blocking buffer with mouse monoclonal anti-mCherry-Tag antibody (1:1000; St Jonh's Laboratory #STJ34373), rabbit polyclonal anti-flag (1:1000; Sigma # F7425). Washes were performed using PBS-T and secondary incubation was performed for 1 h at room temperature with goat anti-mouse Alexa Fluor Plus 800 (1:5000; Thermo Fisher Scientific, #A32730), goat anti-rabbit StarBright Blue 700 (1:5000; Bio-Rad, #12004161), human Fab anti-actin rhodamine-conjugated (1:10000; Bio-Rad, #12004164). Immune complexes were visualized using the ChemiDoc MP Imaging System and analyzed using Bio-Rad Image Lab (version 6.1).

**U6 promoter evaluation.** CRISPR KO efficiency of the stable mCherry reporter integrated into the mosquito cell lines was used as a readout to test relative U6 promoter sgRNA expression of different mosquito promoters, using transient transfection. Plasmid DNA for transfections was prepared using mini or midiprep kits (Qiagen) and quality was analyzed by spectrophotometry and agarose gel electrophoresis. Plasmid DNA concentration was measured (Qubit, ThermoFisher) and plasmid mixes for transfection were normalized by copy number according to the M.W. of each plasmid and using pUC19 (ThermoFisher) to normalize total DNA amounts as needed. All cells were transfected with 300 ng of plasmid mix (12 well format) using Effectene (Qiagen) and following the manufacturer's protocol. The percentage composition of each plasmid mixture used for transfection was as follows: for the *Anopheles* and *Culex* cell lines, a plasmid mixture containing 150 ng (50%) of pAct::Cas9-2A-Neo, and 150 ng (50%) of one of 14 different copy-number balanced mCherry-sgRNA expressing plasmids was transfected. Additionally, two control transfections were performed: one including the "empty" pLib6.4-Agam_695 expression vector (sgControl), that effectively drives the expression of a "non-targeting guide" matching the sequence of the empty BbsI cloning cassette, and a second including the same control guide but lacking the Cas9-expressing plasmid. Transfections in the *Aedes* cell line were performed with a different plasmid mixture containing 150 ng (50%) of pAaePUb::Cas9-2A-Neo as a source of Cas9 and 135 ng (45%) of one of 14 different copy-number balanced mCherry-sgRNA expressing plasmids and, in addition, 15 ng (5%) of pSL1180-HR-PUbEGFP-NLS to increase fluorescence output of the GFP reporter. For this cell line, we used the same sgControl plasmids, transfected with or without the Cas9-expressing plasmid. Transfections in the *Drosophila* cell line were performed with 300 ng (100%) of each one of 14 different copy-number balanced mCherry-sgRNA expressing plasmids and sgControl plasmid as a control. For this experiment since Cas9 was already expressed in the cell line transfection with Cas9 expressing

plasmid was not necessary. Additional information on plasmid vectors used is provided in Supplementary Data 1. Transfections were performed in 12-well format with 3 or more replicate wells per condition tested, and a total of three independent experiments were performed. *Culex* and *Ae. albopictus* cells were detached from flasks using Accumax (Innovative Cell Technologies, Inc) and seeded onto 12-well plates 16–24 h before transfection. *Anopheles* cells were resuspended from flasks by pipetting and seeded 30 m before transfection. *Culex* and *Aedes* cell lines were transfected at ~70% confluency, and *Anopheles* cells at ~90% confluency. 24 h after transfection, cells were transferred to new plates and cultured for 12 days, then either analyzed by flow cytometry or slowly frozen at –80 °C in culture media supplemented with 10% DMSO (v/v) and stored until flow cytometry analysis. *Anopheles* and *Aedes* cells during the last passage preceding the experiment end-point were treated with a sub-lethal dose of puromycin (0.5 ug/ml) to increase the ratio of transfected cells and reduce the volume of cells analyzed with flow cytometry (puromycin resistance is conferred by the sgRNA expression plasmid).

**Flow cytometry analysis**. Flow cytometry analysis was performed on a FAC-Symphony analyzer (BD) (Harvard Medical School, Immunology Flow Core). Immediately prior to analysis, frozen cells were thawed quickly in a 30 °C water bath, washed twice in PBS, and resuspended in fresh culture media. Cells were loaded into 96-well plates and read using the high-throughput sampler. Flow speed, laser voltages, and gating parameters were adjusted for each cell line. The gating strategy is shown in Supplementary Fig. 2a. Gates for cell singlets were defined based on forward and side scattering beams and a subordinate gate for GFP$^+$ cells was defined based on untransfected control cells. GFP$^+$ cells were sub-gated to define mCherry$^+$ cells and mCherry$^-$ cells within the GFP population were defined by exclusion from the mCherry$^+$ gate (NOT Boolean gate for mCherry$^+$). The CRISPR knockout efficiency relative to each U6 promoter tested was inferred using the populations defined above as: KO efficiency % = {[# of mCherry$^-$ cells] ÷ [# of total GFP-positive cells]}*100. We also measured the median fluorescence intensity (MFI) of the mCherry signal within the GFP population as an alternative method to quantify KO of the mCherry reporter. MFI is a parameter of the population that correlates with the number of molecules present in each cell, and thus can also be used to quantify KO of a fluorescent reporter. All analysis were performed using FlowJo version 10.7.1 (BD). To adjust for differences in the number of cells analyzed for each sample within a replicate experiment, cells defined by the GFP gate were down-sampled to a fixed arbitrary number using the DownSample plugin. Samples with < 1000 GFP-positive cells were discarded from the analysis. All raw data, source data, and statistics relative to flow cytometry analysis are provided in Supplementary Data 2 and **Source Data** files.

**Rho1 GTPase targeting**. To target the Rho1 GTPase locus (AGAP005160) in Sua-5B-IE8-Act::Cas9-2A-Neo cells, we seeded ~1.8 × 106 cells in 12-well dishes for 30 m and transfected using Effectene (Qiagen) and following the manufacturer's instructions. Cells were transfected with a plasmid mixture containing 150 ng (50%) of pBS130 (Addgene #26290) encoding HSP70-ΦC31-Integrase and 150 ng (50%) of either pLib6.4-Agam_695 "empty" (sgControl) or a sgRNA targeting Rho1 (sgRho1), and then cultured for 7 days in 4.5 μg/ml puromycin selective media. Cells were transferred to 96-well plates (Falcon #353219) and images of control or Rho1 sgRNA-treated cells were captured with a IN Cell Analyzer 6000 confocal imaging system (GE) using a 20X objective and dsRed acquisition settings to record mCherry signal stably reported from the cell line. Measurements of cell area for quantification of the Rho1 phenotype were performed using one field of each control or Rho1 sgRNA-treated cells using Fiji image analysis software with built-in alghorithms (open source ImageJ2, version 2.1.0/1.53c). Briefly, images were converted into 8-bit binary images; thresholding based on the mCherry signal was performed using the "Li" algorithm, followed by "fill holes" and "watershed" processing algorithms, then the number of cells and associated surface area were calculated. Raw cell area measurements are reported in the Source Data file. We used the T7-endonuclease I assay to verify editing at the Rho1 gRNA targeted region. First, genomic DNA was isolated using a Quick DNA Miniprep Kit (Zymo Research) according to the manufacturer's instructions. Phusion polymerase (New England Biolabs) using HF buffer with primers 22-23 was used to obtain a PCR amplicon (877 bp) asymmetrically spanning the genomic target site of the sgRNA on the Rho1 GTPase locus (AGAP005160). The PCR reaction was directly denatured at 95 °C for 5' and slowly cooled down to room temperature to allow re-annealing. Next, 8 ul of the resulting PCR reaction was combined with 1 μl NEB2 buffer and 1 μl T7 Endonuclease I (New England Biolabs), digested 15' at 37 °C, then resolved on a 3% TBE agarose gel at 160 V for 1.5 h.

**Library design, and cloning**. sgRNAs were chosen using CRISPR GuideXpress. Briefly, a list of five *Drosophila* genes (*FKBP12*, *EcR*, *usp*, *Oatp74D*, and *PTP-ER*) were inputted and all computed sgRNAs were retrieved. Additionally, the second set of *Drosophila* genes from a previous study[65] was inputted, and the top six sgRNAs per gene were selected based on the following criteria: minimal OTE (off-target effect) score; maximum ML (machine learning efficiency) score; BbsI-site bearing sgRNAs were culled from batch results. sgRNA sequences were placed within a 109-mer using specific tags and retrieved using dial-out PCR[76]. Each dial-out product was digested with BbsI and separated on a 20% non-denaturing

polyacrylamide TBE gel (ThermoFisher Scientific). The product was next extracted using the crush-soak method and ligated into BbsI-digested pLib6.4-Agam_695. The product was then transformed into *E. cloni* 10GF' ELITE Electrocompetent Cells (Lucigen) and plated onto ten 15 cm LB plates containing carbenicillin and grown overnight at 30 °C. A total of 100 bacterial colonies per construct were harvested into 25 mL of LB medium, mixed with an equal volume of 50% glycerol, and stored in 1 mL aliquots at –80 °C. Prior to transfection, the plasmid library was prepared by miniprep (Qiagen).

**Chemical-genetic CRISPR screening**. Sua-5B-IE8-Act::Cas9-2A-Neo cells in log phase of growth were seeded at $6 \times 10^6$ cells per 6-well dish in growth media containing antibiotics. They were transfected with a plasmid mixture containing equimolar amounts of HSP70-ΦC31-Integrase plasmid (pBS130) and sgRNA donor plasmid library (pLib6.4-Agam_695) using Effectene (Qiagen) according to the manufacturer's base protocol ("1:25"). Transfection efficiency was found to be 22–28% under these conditions, whereas the % of stably recombined cells after one month of passaging without selection was found to be ~3% (Fig. 2c). To achieve ~150 cells per sgRNA, 3487 sgRNAs × 100 cells/sgRNA ÷ 0.03 = ~18 × 10$^6$ were transfected in three wells of a 6-well dish. After 4 days, each well was expanded into a 100 cm dish containing 5 μg/mL puromycin. Cells were cultured for an additional 12 days with media changes every 4 days. Cells were transferred to media containing puromycin and selective drugs, determined to be around the IC50 for Sua-5B-IE8-Act::Cas9-2A-Neo cells: rapamycin (LC Laboratories) was dissolved in DMSO and used at a final concentration of 40 nM; 20-hydroxyecdysone (Sigma) was dissolved in ethanol and used at a final concentration of 50 ng/mL; trametinib (Sellek Chemical) was dissolved in DMSO and used at a final concentration of 400 nM. Cells were continually passaged in the selective medium for 30 days with media changes or re-seeding. Re-seeding density was maintained above 1000 cells/sgRNA at all times. Following selection, cell pellets representing > 1000 cells/sgRNA were extracted using the Quick-gDNA MiniPrep kit (Zymo). Next, the genomic DNA was subjected to 2-step PCR to introduce in-line barcodes, a variable fingerprint, and Illumina sequencing primer and adapters. Amplicons were subjected to sequencing using a NextSeq500 at the Harvard Biopolymers Facility at Harvard Medical School. Computational barcode removal was performed using in-house scripts. Low-read sgRNAs (those with fewer than 10 reads in the plasmid library) were removed from the read count files. All subsequent read count and data analysis were performed using MaGeCK 0.5.7.

**Orthology mapping, sgRNA design, and variant analysis**. *Ortholog mapping between Drosophila melanogaster (Dmel) and Anopheles gambiae (Agam)*. Ortholog mapping was extracted from the following five different prediction algorithms: orthoMCL vs5, eggNOG vs5.0, InParanoid vs8, orthoFinder (for which the code was run locally using the protein reference of Agam from VectorBase and Dmel from RefSeq), and TreeFam vs9. Orthologous pairs were integrated using the same pipeline as DIOPT[42] and each orthologous pair was assigned a DIOPT score, i.e., the count of integrated algorithms that predict the pair (maximum score of 5).

*Ortholog mapping among mosquito species*. Ortholog mapping was obtained using Biomart at Vectorbase (https://biomart.vectorbase.org/biomart/martview). The Biomart ortholog mapping used for this study was subsequently replaced when Vectorbase merged with EuPathDB to become VEuPathDB. Ortholog mapping is now available from the new user interface using a gene ID search strategy with an ortholog transform step (https://vectorbase.org/vectorbase/app/query-grid).

*sgRNA design pipeline*. Supported species and the information about reference genome versions are available at the dropdown menu on the CRISPR GuideXpress search page (https://www.flyrnai.org/tools/fly2mosquito/web/species). The species currently supported at CRISPR GuideXpress and their corresponding genome versions are as follows: *Anopheles gambiae* (AgamP4.12), *Anopheles coluzzii* (AcolM1.8), *Anopheles stephensi* (AsteS1.7), *Aedes aegypti* (AaegL5.2), *Aedes albopictus* (AaloF1.2 & C6/36), and *Culex quinquefasciatus* (CpipJ2.4). Genome sequence and annotation files were obtained from https://vectorbase.org/vectorbase/app/downloads/Pre-VEuPathDB%20VectorBase%20files/. Input files used by the pipeline include

```
<species_name>_BASEFEATURES_<species_version>.gtf,
<species_name>_TRANSCRIPTS_<species_version>.fa, and
<species_name>_CHROMOSOMES_<species_version>.fa.
```

The pipeline starts by using genome annotation to determine the transcript with the longest coding sequence (CDS) for each gene. In cases where there is a tie, the first transcript is chosen arbitrarily. Next, all potential sgRNA designs within the CDS of the selected transcripts are identified and logged. Then, unique k-mers within the genome are identified for computing seed scores for each design. Design sequences are BLASTed against the genome sequence and off-target scores are assigned based on off-target alignment hits with varying numbers of mismatched nucleotides. Off-target hits with fewer mismatches are weighted more heavily. The OTE (Off-target effect) score was calculated based on the number of potential off-target sites at 3 different thresholds and defined as follows: "OTE score = a.bc" where (a) is added to the digit before the decimal point and is the number of off-target sites of the least stringent threshold (only considering off-target sites with 3 or fewer mismatches); (b) is added to the digit after the decimal point and is the number of off-target sites at the moderate threshold (only considering off-target sites with 4 or fewer mismatches) (c) is added as the second digit after the decimal

place and is the number of off-target sites under the most stringent threshold (5 or fewer mismatches). The categorization of sgRNA designs in Fig. 1 b, c, e was defined as follows: "no OTE" (least stringent criteria) = score < 1; with OTE (least stringent criteria) = score ≥ 1.

For each sgRNA, a Housden efficiency score[43] is computed using a position matrix, and a machine learning-based efficiency score is computed using the pipeline available here: https://github.com/PierreMkt/Dmel-sgRNA-Efficiency-Prediction, which is based on *Drosophila* cell CRISPR screens[40]. Finally, a comparison with SNPs associated with genome data from wild populations is calculated based on the Ag1000G dataset[45]: https://www.malariagen.net/data/ag1000g-phase-2-ar1 (available for AgamP4.12 and AcolM1.8). The sgRNA pipeline was written in Python 3 and Perl 5.24.0, and uses BLAST 2.6.0.

*Analysis of wild populations variants.* Efficiency of sgRNAs for *An. gambiae* and *An. coluzzii* in wild populations was evaluated based on the datasets from *Anopheles gambiae* 1000 genomes project[45,46] (Ag1000G) (https://www.malariagen.net/data/ag1000g-phase-2-ar1). Percent efficiency in wild populations was calculated based on the percent of samples carrying the SNP in each of the sgRNA target sequence. The SNP analysis results were obtained from ftp://ngs.sanger.ac.uk/production/ag1000g/phase2/cas9_targets/ [77]while the sample metadata was obtained from ftp://ngs.sanger.ac.uk/production/ag1000g/phase2/AR1/samples/[45].

*Genome variants in Sua-5B cell line.* The Sua-5B cell line was sequenced by BGI Genomics (https://www.bgi.com/global/) and processed as follows. The paired-end sequencing data was QC'd using fastqc 0.11.8 and multiqc 1.9. Pre-processing was performed according to the Broad Institute's best practices workflow for data preprocessing for variant discovery (https://gatk.broadinstitute.org/hc/en-us/articles/360035535912-Data-pre-processing-for-variant-discovery). First, the raw reads were aligned to the *Anopheles gambiae* (AgamP4) genome sequence or separately, to the *Anopheles coluzzii* (AcolM1.8) genome sequence using the Burrows-Wheeler Aligner BWA-MEM algorithm (http://bio-bwa.sourceforge.net/). Next, MarkDuplicatesSpark from the Broad's Genome Analysis Toolkit Genome (GATK) was used to identify read pairs that likely originated from the same original DNA fragments due to artifacts (https://gatk.broadinstitute.org/hc/en-us/articles/360050814112-MarkDuplicatesSpark). Variants were identified by following the Broad Institute's best practices workflow for germline short variant discovery (https://gatk.broadinstitute.org/hc/en-us/articles/360035535932-Germline-short-variant-discovery-SNPs-Indels-). Variant calling was performed using the GATK HaplotypeCaller (https://gatk.broadinstitute.org/hc/en-us/articles/360037225632-HaplotypeCaller). Finally, hard filtering was performed in accordance with the Broad Institute's recommendations for generic hard filtering (https://gatk.broadinstitute.org/hc/en-us/articles/360035890471-Hard-filtering-germline-short-variants). GATK version 4.1.8.1 was used for the analysis.

To obtain RNA-seq expression data from Sua-5B cells, the raw sequencing files were first obtained from VectorBase (https://vectorbase.org/vectorbase/app/record/dataset/DS_1c16f776df), QC'd using fastqc 0.11.8 and multiqc 1.9, and analyzed using Salmon 1.3.0 (https://combine-lab.github.io/salmon/) and tximport 1.16.0 (https://github.com/mikelove/tximport).

### Implementation and user interface (UI) of CRISPR GuideXpress. *Implementation.* CRISPR GuideXpress is available at https://www.flyrnai.org/tools/fly2mosquito. The backend is written in PHP using the Symfony framework and precomputed results are stored in a MySQL database. The front end uses the Twig template engine, Bootstrap, and some custom CSS for the UI. The JBrowse genome browser is used to display results visually and JQuery with a DataTables plugin is used for result tables. The website is hosted by the Research Computing group at Harvard Medical School.

*UI features.* The UI provides three major functions. First, the resource supports ortholog mapping of lists genes between *Drosophila* and *An. gambiae* and among supported mosquito species, including a ranking when one input gene maps to multiple genes. Second, the resource supports searches for sgRNA designs for a single gene or a list of genes. Both specificity scores, including the seed score and OTE score, and predicted efficiency scores are provided, allowing users to select optimal designs from multiple target sites. In addition, genome variants in *Anopheles coluzzii* Sua-5B cells were annotated for each sgRNA design, so that users can avoid target sites that might fail to work in this cell line. Each sgRNA design is also annotated with regards to SNPs as compared with the Ag1000 wild population genome data. The resource also supports the search of expression levels in the *Anopheles coluzzii* Sua-5B cell line, starting with a single gene or a list of genes.

### Additional software. Figures and graphical elements in this manuscript were created and assembled with BioRender (BioRender 2021) and Adobe Illustrator (version 25.2.1). Statistical analysis was performed and plots were drawn with GraphPad Prism (version 9.1.2).

### Reporting summary. Further information on research design is available in the Nature Research Reporting Summary linked to this article.

## Data availability
The raw sequencing reads and VCF files of whole-genome sequencing data from Sua-5B-IE8-Act::Cas9-2A-Neo cell line are available for download at CRISPR GuideXpress (https://www.flyrnai.org/tools/fly2mosquito/web/download) as well as the pilot CRISPR screen result. Source data are provided with this paper. All plasmids and cell lines created in this work are available in repositories listed in Supplementary Data 1. Source data are provided with this paper.

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

## Acknowledgements

We thank Flaminia Catteruccia and Nelson Lau for cell lines. We thank Daniela Silva-Ayala for helpful discussions. The research reported in this paper was supported by NIH NIGMS grant P41GM132087 to N.P. N.P. is an investigator of Howard Hughes Medical Institute.

## Author contributions

RV and EM contributed equally to this work. RV, EM, YH, SEM and NP conceived the project. RV, EM, YH, SEM, TMC and NP contributed to the design of the experiments. JR, PM and YH developed the bioinformatic tools. RV, EM and FFS performed the experiments and contributed to the collection and analysis of data. EM wrote the first draft of the paper with input from RV, SEM and NP. All authors edited and approved the paper.

## Competing interests

The authors declare no competing interests.
