## [Peer Review File · Nature Communications]

Reviewers' Comments:

Reviewer #1:

Remarks to the Author:

Viswanatha and colleagues describe a set of molecular and bioinformatic tools for performing high-throughput CRISPR screens in mosquito cell cultures, with the complete pipeline for generating CRISPR knockouts validated in *Anopheles coluzzii* cell cultures. Well-developed tools for performing high-throughput CRISPR screens in mosquito cell cultures have been lacking, and the tools reported here will transform the field by enabling functional studies into pathogen-host interactions and the molecular biology of mosquito cells. The online bioinformatic resource (CRISPR GuideXpress) seems to have been made publicly available already and will greatly facilitate sgRNA design not just for high-throughput studies but also targeted CRISPR studies, as have been performed already by a number of research groups. The online resource is particularly useful as it includes a number of commonly studied mosquito species, for which promoters for sgRNA expression were validated in this study.

My major comment regards the generation of the mCherry cell lines, where some clarification is needed to fully interpret the data presented in Fig 1 and Supplementary Fig 1 and 2. The authors state that "integrated targets [like the mCherry construct used here] are expected to reproduce gene repair outcomes with the same dynamics as native genes". Is there evidence (from the authors or other labs) to confirm this is the case? How many copies of the mCherry sequence are integrated into each of the mosquito cell lines? This is important to know so that the knockdown efficiency with different U6 promoter constructs (Fig 1) can be interpreted, since a difference in mCherry copy number might affect the data obtained from the different cell lines. It would also be useful to have this information so as to determine whether the data reflect anticipated knockout efficiencies for the two alleles one would expect to be present for endogenous genes. In the panel to Supplementary Fig 1, the authors state that "In rare cases (~0.1%), this successfully results in expression of mCherry as a protein fusion". Do the authors have any data to substantiate this statement? Supplementary Fig 1b, 1c and 1d need to have negative controls (lacking mCherry integration) included to allow proper evaluation of the mCherry expression in the transgenic cells. Do the authors have an explanation for why the mCherry localisation, particularly in C6/36 cells, is quite different between the images? What do the different images represent, different clones? If so, which clone was selected for the follow-on experiments?

For the experiments in Fig 1, where U6 promoters were validated in different mosquito cell lines, what was the transfection efficiency for the GFP/U6 plasmids and were there differences in the expression levels of the GFP? These data would be important to include so that the differences in CRISPR efficiency can be better interpreted, since the data would give an indication as to whether each construct is similarly expressed in each mosquito cell type.

Minor Comments

Supplemental Fig 1 could be labelled better to explain what the authors mean by "Exn", "A", "D", "attP" etc. in panel 1A.

"Up to three" U6 promoters were chosen for each mosquito species for functional validation, how were the promoters selected in cases where more were available?

Since the authors include some cell line reference sequences in their online sgRNA design tool, why was the Aag2 reference sequence not included for *Aedes aegypti*? This is perhaps the most commonly studied *Aedes* cell line and its inclusion in the resource would greatly enhance its utility.

Reviewer #2:

Remarks to the Author:

Summary

The authors use a RMCE system to deliver multiple sgRNA targeting specific genes into mosquito cell lines and evaluate their capacity to achieve gene knockout. They test the system in different cell lines derived from *Anopheles*, *Culex* and *Aedes*. This is a useful resource and I like that they developed it for all the main mosquito vector species. However, application of RMCE in insect cells is not a novel technique (i.e. Mannivanan, 2015) and results obtained in the generation and application of the lines should be explained in further depth in the main text (see substantive points to consider). The authors validate the functionality of native and orthologous U6 promoters from four mosquito species, as well as generating a mosquito consensus U6 promoter that works well in all three tested lines. mCherry-specific sgRNA driven by different U6s are used to KO the previously inserted mCherry in each of the lines and infer U6 functionality by the lack of Red in the cells. The understanding and capacity to use different U6 promoters is needed in insect genetic engineering, especially in a gRNA-multiplexing scenario (i.e. Oberhofer 2018, and others) where one would use different U6s to avoid self-recombination of the transgene as much as possible. I really like the approach that the authors took in this section as well as the main and supplementary figures associated with the experiments. In the discussion of the results (which lacks detailed and specific information), the authors should hypothesize why all U6s work particularly well in *Culex*, contrary to what occurs in *Anopheles/Aedes*, which is more in accordance with what we constantly observe *in vivo*. In a recent paper published in Nat Commun and co-authored by the same group (Feng 2021), Cquinq_801 (U6:4) does not display any KO efficiency in Hsu cells, whereas in this study it KOs 30% of the targets in the exact same cell line. This discrepancy is worth assessing or explaining. Another problem not discussed is how to further extrapolate cell data. Testing U6s that work well in many species/cell lines *in vivo* would be interesting, as we know that U6s can work well if the genomes are not distant (Xu, 2019; where Dgrim works in a Dmel background). To my knowledge, not many non-native U6s have worked *in vivo* in mosquitoes so, while the experiments are costly, they would provide the paper with a big boost in impact and relevance. In between the two cell-based assays, the authors describe GuideXpress, a resource they developed to predict orthologs between *Drosophila* and multiple mosquito species and generation of sgRNAs. While, again, this is a good resource to have in the community, it does not flow well (as it stands) in the paper and I wonder whether it should be included there. The authors did not use the tool to predict the U6 orthologs used earlier (they use Vectorbase's own BLAST interface and multiple alignment), but they use it to design sgRNAs against the genes they test for KO in the last experiment of the paper. Personally, I am not a bioinformatician but a user of such resources and maybe I did not fully grasp the differences between GuideXpress and other similar tools, which most of them already support important vector species and other features described here. I do, however, like that you can retrieve sgRNAs that work for multiple species as well as the inclusion of the 1000 genome data for Agam and Acol. The authors test their cell system in different genes starting with *Rho1*, where they evaluate its KO phenotypically, as well as an editing proxy with gel cleavage assays. This is a good way to test the efficiency of the system, which could be strengthened with sequencing data to confirm specific editing of the cells. In this they use the RMCE lines

to integrate the different sgRNAs, which also contain Cas9. It became confusing to follow, as the addition of stable Cas9 into the system was not explained prior. Figure legend has information on how, but it should also be included in the main text as it is highly relevant for the flow of the article. I would also like to know the efficiency of RMCE conversion in cells or the selection process of sgRNA+ vs sgRNA- cells. Since sgRNAs are marked with GFP to RMCE's mCherry, I assume the authors sorted the cells for fluorescence, but should be explained. The paper concludes with a CRISPR screen on five genes, however the results focus heavily on *FKBP12* and the other four genes are briefly mentioned. It would be beneficial to see Figure 3e applied to all the other genes tested, at least in a supplementary figure (including the Ecl ortholog where they do not see any enrichment). Again, discussion of these results is lacking.

Unfortunately, for the reasons stated above, I do not recommend this paper for publication in Nature Communications. I do not think that the manuscript, as it currently stands, is in accordance with the journal's previous work, novelty requirements and reputation.

I welcome questions or concerns about the content of the review by manuscript authors or editors at Nature Communications. I do not think this is the case, but if my name should not be attached to the revision due to journal's policy, please feel free to delete it to keep it anonymous.

Best,
Gerard Terradas (gkt5113@psu.edu)

Substantive points to consider

1. The readability of the paper could be improved in a significant way to make the study more impactful, accessible and clearer to the reader. Several examples are given in the specific minor points section below, but I found that the manuscript is lacking in clarity in some areas, where key information is hidden in supplementary figures or legends and not in the main text. Also, transition between paragraphs should improve.
2. The manuscript is not structured in sections, which should be done to comply with the Nat Commun format. I believe that it would be highly beneficial to the reader and the quality of the manuscript to expand certain sections: especially the introduction, which is too brief, and addition of the discussion. While some results are described well in the text (Fig.1 – KO efficiencies), some others (Figure 3) are not described with numbers but just with observations.
3. Referencing throughout the text is very minimal for key studies (see below).

Minor points

1. References are necessary in the introduction, in addition to expanding: Added some that I know well from the top of my head, but others left as Ref -- Current efforts to fight malaria and other mosquito-transmitted diseases such as Dengue, Zika, Chikungunya and West Nile Virus rely on control of vector populations, mostly by means of insecticides

(ref). These measures are hampered by ever-increasing insecticide resistance (ref). Alternative strategies under current development include those based on the use of endosymbiotic bacteria (i.e. Walker 2011, Utarini 2021) or gene-drives to suppress wild mosquito populations (i.e. Hammond 2016, Kyrou 2018) or replace them with disease-refractory mosquitos (i.e. Gantz 2015, Adolphi 2020, Carballar 2020) 2.

2. “Gene drives” is correct, instead of using “gene-drives”. Do not hyphenate nouns, only adjectives: i.e. gene drive (noun is ‘drive’) but gene-drive system (noun is ‘system’). Other cases throughout the text: GFP-transfected cells; gRNA-expressing vector; better-annotated genome.
3. The paper is focused in mosquito biology and if you want to add an extra layer of complexity (virus, insecticides) you would be studying the interaction between virus/insecticide and mosquito, not the disease. Studies of mosquito-borne diseases would benefit of the availability of methods that allow large-scale functional cell-based screens, for example genome-scale screens for virus, bacteria or parasite entry, innate immunity, or resistance to insecticides and other toxins.
4. Reference your own work: Previously, we developed such a method in *Drosophila* cells
5. “Strong”, not “strongly”: we selected a single, strongly mCherry-positive
6. The info on which *DmelU6* promoter was used is only present in the methods. I would add in the main text that you used U6-2 as a starting point on the ortholog search. I also have a comment on this and it is to make sure that you used U6-2 (it is not used much in *in vivo* genetic engineering because it displays low expression of the gRNAs, whereas U6-1 and U6-3 are widely used as gRNA promoters). I think that, in a paper where you are trying to find highly expressed U6s, it is worth mentioning and discussing: we first used BLAST and multiple alignment to identify orthologs of the *Drosophila* U6 promoter
7. VEuPathDB references only for AGAP13695, not for the rest of U6s: RNA-seq data suggests they are expressed in cell lines and in adult tissues (see Methods).
8. KO is used throughout the text and it has already been introduced previously as “knockout”, use “KO”: all mosquito promoters tested elicited measurable knockout
9. This refers to 1e, not 1d. Should be 1d given the flow of the text. overall mean KO efficiency of about 30% (Fig. 1d, Supplementary Figure 2b)
10. In this case 1d is correct but should be moved to 1e since the results are shown after the *Culex* ones. with about 27% mean KO efficiency (Fig. 1d, Supplementary Figure 2b).
11. No comma. allowing in some cases, inter-species
12. Where do these cells come from? To this point, there is no mention of their creation, only Sua-5B-IE8. We used the *Anopheles coluzzii* Sua-5B- IE8-Act::Cas9-2A-Neo cell line
13. Citation #15 has no validity here and, unless I am wrong, the data from 9 only elicited PTP-ER as a factor. If the authors write that PTP-ER (and the same occurs for the other genes) is a negative regulator of MAPK, which in turn can be suppressed with trametinib, both claims should be backed up with references. This, as mentioned, is a common problem of the paper. We first chose five genes that had previously been shown to be drug-resistance factors in *Drosophila* cells^{9,15}
14. You did not perform CRISPRa or RNA knockdown screens in the paper. This includes not only application of CRISPR-Cas9 knockout screening but also CRISPR/Cas-based activation and RNA knockdown screens.

15. This is almost copy-pasted from paragraph 2 (see point 3 above). CRISPR pooled screening in these cells can be combined with a variety of cell-based assays, including assays relevant to virus or parasite entry, innate immunity, resistance to insecticides and other toxins

Responses to Reviewer Comments

We thank the reviewers for their comments. As a preface to our point-by-point responses (below), we would like to summarize the major changes we made in this revised version of our manuscript. First, in response to the reviewers' comments, we significantly expanded the "Introduction" and "Discussion" sections. Second, in response to reviewer # 2's suggestion, we changed order of the figure introducing our bioinformatic pipeline. Third, in accordance with the increased space in Nature Communications (we had initially submitted to Nature Methods as a *Brief Communication*, and the manuscript was transferred directly to Nature Communications), we expanded the number of figures from 3 to 5. Fourth, for added clarity regarding the reviewer' questions, we added additional data to demonstrate the recombination-mediated cassette exchange (RMCE) process.

Reviewer #1 (Remarks to the Author):

Viswanatha and colleagues describe a set of molecular and bioinformatic tools for performing high-throughput CRISPR screens in mosquito cell cultures, with the complete pipeline for generating CRISPR knockouts validated in *Anopheles coluzzii* cell cultures. Well-developed tools for performing high-throughput CRISPR screens in mosquito cell cultures have been lacking, and the tools reported here will transform the field by enabling functional studies into pathogen-host interactions and the molecular biology of mosquito cells. The online bioinformatic resource (CRISPR GuideXpress) seems to have been made publicly available already and will greatly facilitate sgRNA design not just for high-throughput studies but also targeted CRISPR studies, as have been performed already by a number of research groups. The online resource is particularly useful as it includes a number of commonly studied mosquito species, for which promoters for sgRNA expression were validated in this study.

We thank the reviewer for their summary and kind words about our work.

My major comment regards the generation of the mCherry cell lines, where some clarification is needed to fully interpret the data presented in Fig 1 and Supplementary Fig 1 and 2. The authors state that "integrated targets [like the mCherry construct used here] are expected to reproduce gene repair outcomes with the same dynamics as native genes". Is there evidence (from the authors or other labs) to confirm this is the case?

This method, called "MiMIC RMCE" for Minos-mobilized integration cassette-recombination mediated cassette exchange, has been used previously in *Drosophila* to tag endogenous genes while maintaining their native transcriptional and post-transcriptional regulation. Nearly 8,000 MiMIC insertion fly lines have been constructed and have enabled numerous studies to report the localization or expression pattern of the trapped gene (PMID: 29565247). Any repair mechanism that happens in the genome will happen with the MiMIC cassette inserted into the genomic locus because it is part of the genetic locus and becomes spliced into the native transcript. For clarification, we now phrase the sentence as follows: "An improvement of this approach over a plasmid-based dual reporter assay is that mCherry is genomically encoded rather than episomal target, revealing repair outcomes that would be expected at a native gene." (Lines 180-182).

How many copies of the mCherry sequence are integrated into each of the mosquito cell lines? This is important to know so that the knockdown efficiency with different U6 promoter constructs (Fig 1) can be interpreted, since a difference in mCherry copy number might affect the data obtained from the different cell lines. It would also be useful to have this information so as to determine whether the data reflect anticipated knockout efficiencies for the two alleles one would expect to be present for endogenous genes.

The purpose of our U6 promoter evaluation was to find the most effective promoter for each cell-line, not to compare across cell-lines. While as observed from the reviewer, insertions into multiple loci may reduce the observed cutting efficiency, the relative efficiencies of the promoters would still be evident in our analysis because this is comparison within the same cell line. We now add this clarifying sentence:

“Although the number of insertions and expression level of mCherry vary between cell lines, this approach permits comparison of different U6 promoters within the same cell line.” (Lines 182-184). In the case of the *Anopheles* Sua-5B-IE8 cell line, we now know from an analysis of our whole genome sequence (WGS) from this cell line that 5 different mCherry cassettes have been integrated in the genome (Supplementary Figure 1e), although this likely resulted in a single mCherry-expressing protein as seen by anti-mCherry western blot (Figure 2b).

In the panel to Supplementary Fig 1, the authors state that “In rare cases (~0.1%), this successfully results in expression of mCherry as a protein fusion”. Do the authors have any data to substantiate this statement?

We have now provided a western blot showing the mCherry fusion protein from the Sua-5B-IE8 subline which is absent from the parental line (Figure 2b). The size of the band (~120 kDa) indicates that it is a fusion with an endogenous protein rather than free mCherry.

Supplementary Fig 1b, 1c and 1d need to have negative controls (lacking mCherry integration) included to allow proper evaluation of the mCherry expression in the transgenic cells.

The mCherry signal in each cell line is very bright and discernable above the background for each cell line we show in Supplementary Figure 1b,c,d. Since we did not image the parental cell lines at the same time, it would be difficult to prepare the mCherry negative control without thawing of many frozen cell lines. We hope that as a compromise, the reviewer will accept characterization of the Sua-5B cell line in the updated version of Figure 2b and Supplementary Figure 1e.

Do the authors have an explanation for why the mCherry localisation, particularly in C6/36 cells, is quite different between the images? What do the different images represent, different clones? If so, which clone was selected for the follow-on experiments?

The MiMIC RMCE system works by randomly mobilizing a transposon from a plasmid into the genome at low frequency. The transposon contains a promoterless mCherry flanked by a splice donor and splice acceptor that, when inserted into an intron, becomes a protein fusion with the native gene. This is why each clone produced by this method is expected to have a different mCherry localization, as each one reveals the localization of a different native gene. This method was demonstrated previously in *Drosophila* cells (Neumüller et al., 2012 Genetics). We now add explanatory text as follows: “As expected[50], we observe different mCherry distributions in each clonal isolate (Supplementary Figure 1b-d).” (Line 142-143)

For the U6 promoter optimization, Sua-5B-IE8, C6/36-HE8, and Hsu-1.7 cells were used. For the CRISPR screens, Sua-5B-IE8 was used.

For the experiments in Fig 1, where U6 promoters were validated in different mosquito cell lines, what was the transfection efficiency for the GFP/U6 plasmids

We now added the transfection efficiency as a new column in Supplementary Data 2 for each sample/cell line, namely “Cells/Single Cells/GFP+ | Freq. of Parent”. Transfection efficiency is defined as the proportion of gated cells GFP+ cells over the parent population. Note that this value is not equivalent to the “true” transfection efficiency for two reasons: 1) it was recorded 12 days post transfection, as this is the amount of time we needed to allow in order to detect maximal CRISPR knockout effects. Because the transfected plasmid is continually diluted with successive cell divisions, a truer estimate of transfection efficiency would have been at 24-48 hours post transfection, but this would be too early for KO detection. 2) To counteract the dilution effect of the plasmid transfection (as noted in the Methods section), for *Anopheles* and *Aedes* cell lines the last cell passage was performed in media supplemented with puromycin to increase the proportion of GFP-2A-Puro(R) plasmid transfected cells to maintain a sufficient population of transfected cells for meaningful flow cytometry statistics.

and were there differences in the expression levels of the GFP?

Regarding across-cell-line variation, there were expected differences in GFP expression most likely due to differences in *PoI II* promoter expression strength among different cell lines, e.g. the *Drosophila* Actin5C promoter works better in *Anopheles* and *Culex* but is weaker in *Aedes*. Also as expected, there was relatively little systematic variation within cell lines. The exact GFP intensities for each sample analyzed is presented in **Supplementary Data File 2** for each sample (column "I": "GFP+ | Median (Alexa Fluor 488-A)").

These data would be important to include so that the differences in CRISPR efficiency can be better interpreted, since the data would give an indication as to whether each construct is similarly expressed in each mosquito cell type.

Because the flow cytometry approach we use allows us to include/exclude single cells from the analysis based on desired properties, we gate the cells according to the expression of GFP, a direct measure of the amount of sgRNA expression plasmid the cell received, and then measure the amount of mCherry in this population of cells only. This allows us to overcome issues related to variation in transfection efficiency. We clarified the paragraph in the text as follows: "Mosquito cells with genomically-encoded mCherry allowed us to use a flow cytometry-based dual reporter assay to directly compare KO efficiency in cells expressing the same sgRNA from different pol III promoters (Figure 3b, Supplementary Figure 1, Supplementary Data 1). In this strategy, we test U6 promoter strength by measuring the ability of the downstream sgRNA to suppress mCherry. Specifically, we co-transfected mCherry expressing cells with a Cas9 expression vector and a plasmid containing an mCherry-targeting sgRNA driven by a variable U6 promoter. The U6 promoter plasmid co-expresses GFP as an indicator of transfection. After gating cells with GFP expression, the ratio of mCherry- cells is used to determine KO efficiency." (**Lines 172-179**)

Minor Comments

Supplemental Fig 1 could be labelled better to explain what the authors mean by "Exn", "A", "D", "attP" etc. in panel 1A.

We have now modified the figure to spell out "exon" and provided a legend as an inlay in the figure that defines "A" (splice aceptor site) and "D" (splice donor site). (**Supplementary Figure 1a**)

"Up to three" U6 promoters were chosen for each mosquito species for functional validation, how were the promoters selected in cases where more were available?

We focused on promoters that 1) contain bipartite conserved elements and 2) have some evidence of expression. Evidence of expression came from public RNAseq data from either cell lines or adult tissues readily accessible as added "tracks" on Vectorbase section of VEuPathDB. We now change the text to make this more clear as follows: "To choose promoters, we first used BLAST and multiple sequence alignment to identify orthologs of the *Drosophila* U6-2 (snRNA:U6:96Ab) promoter and chose eleven orthologous promoters from U6 snRNAs of *Anopheles*, *Culex*, or *Aedes* (Figure 3a). When possible, we selected a minimum of three promoters per species, prioritizing U6 promoters that contain an intact pol III bipartite promoter motif and for which RNA-seq data suggests they are expressed in cell lines and in adult tissues (see Methods)." (**Lines 162-168**)

Since the authors include some cell line reference sequences in their online sgRNA design tool, why was the Aag2 reference sequence not included for *Aedes aegypti*? This is perhaps the most commonly studied *Aedes* cell line and its inclusion in the resource would greatly enhance its utility.

We agree that this resource would be of great interest to the community and thank the author for suggesting it! There is currently a genome assembly for Aag2, but to our knowledge lacks gene annotation. Therefore, the most efficient strategy would be to use the most recent existing *Aedes aegypti*

assembly, AaegL5.2 (currently included in GuideXpress), and then alert the user to SNPs in pre-computed sgRNAs between the *Aedes* assembly and the Aag2 genome sequence, as we did for Sua-5B cells. This will be implemented shortly by the bioinformatics team.

Reviewer #2 (Remarks to the Author):

Summary

The authors use a RMCE system to deliver multiple sgRNA targeting specific genes into mosquito cell lines and evaluate their capacity to achieve gene knockout. They test the system in different cell lines derived from *Anopheles*, *Culex* and *Aedes*. This is a useful resource and I like that they developed it for all the main mosquito vector species. However, application of RMCE in insect cells is not a novel technique (i.e. Mannivanan, 2015)

We appreciate that RMCE is not a novel technique to the mosquito community, but its use as a landing site to deliver a controlled and small number of DNA elements to cells in order to perform barcoded screens is a novel use.

and results obtained in the generation and application of the lines should be explained in further depth in the main text (see substantive points to consider). The authors validate the functionality of native and orthologous U6 promoters from four mosquito species, as well as generating a mosquito consensus U6 promoter that works well in all three tested lines. mCherry-specific sgRNA driven by different U6s are used to KO the previously inserted mCherry in each of the lines and infer U6 functionality by the lack of Red in the cells. The understanding and capacity to use different U6 promoters is needed in insect genetic engineering, especially in a gRNA-multiplexing scenario (i.e. Oberhofer 2018, and others) where one would use different U6s to avoid self-recombination of the transgene as much as possible. I really like the approach that the authors took in this section as well as the main and supplementary figures associated with the experiments.

We thank the reviewer for these kind comments and pointing out another potential use of having multiple high-expression U6 promoters for the same species. We now include the Oberhofer reference in the discussion. "Furthermore, having multiple U6 promoters enables combinatorial expression of sgRNAs in the same cell, reducing the chances of recombination between identical U6 promoter sequences^{62,63}." (Line 320-322)

In the discussion of the results (which lacks detailed and specific information), the authors should hypothesize why all U6s work particularly well in *Culex*, contrary to what occurs in *Anopheles/Aedes*, which is more in accordance with what we constantly observe in vivo.

Based on our reading of the literature, we do not believe there is currently enough knowledge about the evolutionary constraints on *PoI III* promoter expression to speculate on why this is so.

In a recent paper published in Nat Commun and co-authored by the same group (Feng 2021), Cquinq_801 (U6:4) does not display any KO efficiency in Hsu cells, whereas in this study it KOs 30% of the targets in the exact same cell line. This discrepancy is worth assessing or explaining.

We did indeed notice this discrepancy, and suggest it is due to differences in the exact promoter sequences used. The vector in Feng et al, constructed in the Gantz lab, includes 1253 bp upstream of the transcription start site (TSS) and was obtained by PCR from genomic DNA of adult mosquitoes. Consistent with how all of the U6 promoters were cloned in this study, we synthesized 171 bp upstream of the TSS based on the reference genome, the region containing the minimal bipartite promoter elements. Besides differences in the overall length between the two constructs, we also identified differences in positions -3 and -7 from the TSS that could also affect expression. The exact sequence used in this study is provided in Supplemental Data File 2.

Another problem not discussed is how to further extrapolate cell data. Testing U6s that work well in many species/cell lines *in vivo* would be interesting, as we know that U6s can work well if the genomes are not distant (Xu, 2019; where Dgrim works in a Dmel background). To my knowledge, not many non-native U6s have worked *in vivo* in mosquitoes so, while the experiments are costly, they would provide the paper with a big boost in impact and relevance.

The U6 promoter evaluation was largely directed towards providing tools for *in vitro* pooled screening in mosquito cell lines. While we share the enthusiasm of the reviewer for pursuing *in vivo* experiments based on our data, we believe that it is outside the scope of this work.

Related to interspecies differences in heterologous U6 promoter usage, we now provide data showing that mosquito U6 promoters tend to work surprisingly efficiently in *Drosophila* cells (**Supplementary Figure 2c**). Conversely, previous studies as well as our own data (**Figure 3**) have shown that *Drosophila* U6 promoters do not work well in mosquito cells. This unexpected result reiterates the reviewer's point that a future project of great interest is to systematically expand this approach outside of mosquitos and test heterologous promoters amongst various dipteran species to identify the evolutionary constraints on *Pol III* promoters in dipterans.

In between the two cell-based assays, the authors describe GuideXpress, a resource they developed to predict orthologs between *Drosophila* and multiple mosquito species and generation of sgRNAs. While, again, this is a good resource to have in the community, it does not flow well (as it stands) in the paper and I wonder whether it should be included there.

We agree with the reviewer's excellent point! According to the reviewer's suggestion, we have now reorganized the paper and present the bioinformatic resource first, then the creation of the RMCE cell line, then the U6 promoter optimization, then the screen. We feel this arrangement flows better and thank the reviewer for this very constructive comment.

The authors did not use the tool to predict the U6 orthologs used earlier (they use Vectorbase's own BLAST interface and multiple alignment), but they use it to design sgRNAs against the genes they test for KO in the last experiment of the paper.

CRISPR GuideXpress currently only works for protein-coding genes, as these are the possible targets of the CRISPR knockout approach. It cannot be used to predict orthologs of non-coding genes such as U6-snrRNAs. We emphasized this point by adding this text: "To facilitate CRISPR-based genome engineering in mosquitos and provide a batch-mode design resource for pooled CRISPR knockout screening **targeting protein-coding genes**, we developed a new online resource for mosquitos, CRISPR GuideXpress (<https://www.flyrnai.org/tools/fly2mosquito/web/>), which has a number of features." (**Line 100-103**)

Personally, I am not a bioinformatician but a user of such resources and maybe I did not fully grasp the differences between GuideXpress and other similar tools, which most of them already support important vector species and other features described here. I do, however, like that you can retrieve sgRNAs that work for multiple species as well as the inclusion of the 1000 genome data for Agam and Acol.

CRISPR GuideXpress was initially developed to support pooled CRISPR screens, and it has features that are not found in other tools for mosquitos. As we state in the results section (**lines 98-124**), key advantages are that GuideXpress 1) permits gene entry and sgRNA output in batch mode, necessary for making a large library; 2) takes input genes from *Drosophila*, which is best annotated so that the user can make testable predictions about the genes that may regulate a function; 3) the sgRNA designs come with pre-computed parameters that can easily be sorted by the user either in the user interface (UI) or in a spreadsheet application to facilitate library construction. 4) offers optimized design of guides with a) cell specific matching data in case of Sua -5B cells; b) guide efficiency data on wild population. Additionally,

we now point out that in the discussion (**lines 302-312**) that it can be used to create focused CRISPR libraries, functionally validating genesets from other studies.

The authors test their cell system in different genes starting with Rho1, where they evaluate its KO phenotypically, as well as an editing proxy with gel cleavage assays. This is a good way to test the efficiency of the system, which could be strengthened with sequencing data to confirm specific editing of the cells. In this they use the RMCE lines to integrate the different sgRNAs, which also contain Cas9. It became confusing to follow, as the addition of stable Cas9 into the system was not explained prior. Figure legend has information on how, but it should also be included in the main text as it is highly relevant for the flow of the article.

We have now included a new **Figure 2** which should guide readers through the process of constructing the CRISPR-ready cell line. Specifically, using Sua-5B cells as an example, we illustrate the steps used to create an RMCE acceptor cell line, adding Cas9 expression, and performing RMCE.

I would also like to know the efficiency of RMCE conversion in cells or the selection process of sgRNA+ vs sgRNA- cells.

1) Regarding the recombination efficiency: Our initial experiments were focused on the screen outcome, so we did not report recombination efficiency. We find that the Φ C31 RCME efficiency is high in *Anopheles* cells (~85%), but the transfection efficiency is low compared with *Drosophila* cells. At the reviewer's suggestion, we have now provided this information (**Figure 2c**). We have also included a lengthy discussion of these results. 2) Regarding efficiency of selecting sgRNA+ cells: Since the sgRNA RMCE cassette also contains a GFP-T2A-Puro(R), we enriched for cells that contain sgRNAs by selecting in 5 ug/mL puromycin. This information is contained in the results (**lines 249-253**) and methods. This results in nearly 100% GFP+, puromycin-resistant cells as expected (not shown).

Since sgRNAs are marked with GFP to RMCE's mCherry, I assume the authors sorted the cells for fluorescence, but should be explained.

We use GFP expression as a visual marker to follow the selection process, but do not use fluorescence sorting to enrich for the sgRNA+ population. Puromycin selection is adequate. This is explained in the Methods section (**lines 631-634**).

The paper concludes with a CRISPR screen on five genes, however the results focus heavily on FKBP12 and the other four genes are briefly mentioned. It would be beneficial to see Figure 3e applied to all the other genes tested, at least in a supplementary figure (including the Ecl ortholog where they do not see any enrichment).

We note that all readcount files from the CRISPR screen were included in a supplementary table for re-analysis (**Supplementary Data File 4**). Also, we had included **Supplementary Figure 3** that showed enrichment of PTP-ER, EcR, and Usp. According to the reviewer's suggestion, we have now reformatted the supplementary file to match the results in the main figure (current **Figure 5**) and included the sgRNAs for the Ecl ortholog.

Again, discussion of these results is lacking.

We expanded the discussion of the results per the reviewer's suggestion.

Unfortunately, for the reasons stated above, I do not recommend this paper for publication in Nature Communications. I do not think that the manuscript, as it currently stands, is in accordance with the journal's previous work, novelty requirements and reputation.

I welcome questions or concerns about the content of the review by manuscript authors or editors at Nature Communications.

Best,
Gerard Terradas (gkt5113@psu.edu)

Substantive points to consider:

1. The readability of the paper could be improved in a significant way to make the study more impactful, accessible and clearer to the reader. Several examples are given in the specific minor points section below, but I found that the manuscript is lacking in clarity in some areas, where key information is hidden in supplementary figures or legends and not in the main text. Also, transition between paragraphs should improve.

We originally formatted this paper as a *Brief Communication* in Nature Methods, and then had it editorially transferred to Nature Communications. We gladly welcomed the opportunity to make the manuscript more readable and adhere to Nature Communications format. We have now moved a number of items out of supplementary figures and included them in the main text and expanded to 5 figures.

2. The manuscript is not structured in sections, which should be done to comply with the Nat Commun format. I believe that it would be highly beneficial to the reader and the quality of the manuscript to expand certain sections: especially the introduction, which is too brief, and addition of the discussion. While some results are described well in the text (Fig.1 – KO efficiencies), some others (Figure 3) are not described with numbers but just with observations.

As mentioned above, we originally formatted this paper as a *Brief Communication* for Nature Methods, and it was editorially transferred to Nature Communications. We have now added section headers complying with the Nature Communications format.

3. Referencing throughout the text is very minimal for key studies (see attached word document for minor points).

We have now included additional references.

Minor points

1. References are necessary in the introduction, in addition to expanding: Added some that I know well from the top of my head, but others left as Ref -- Current efforts to fight malaria and other mosquito-transmitted diseases such as Dengue, Zika, Chikungunya and West Nile Virus rely on control of vector populations, mostly by means of insecticides (*ref*). These measures are hampered by ever-increasing insecticide resistance (*ref*). Alternative strategies under current development include those based on the use of endosymbiotic bacteria (i.e. Walker 2011, Utarini 2021) or gene-drives to suppress wild mosquito populations (i.e. Hammond 2016, Kyrou 2018) or replace them with disease- refractory mosquitos (i.e. Gantz 2015, Adolfi 2020, Carballar 2020) 2.

We thank the reviewer for these references. We have included all of them.

2. “Gene drives” is correct, instead of using “gene-drives”. Do not hyphenate nouns, only adjectives: i.e. gene drive (noun is ‘drive’) but gene-drive system (noun is ‘system’). Other cases throughout the text: GFP-transfected cells; gRNA-expressing vector; better-annotated genome.

Accepted.

3. The paper is focused in mosquito biology and if you want to add an extra layer of complexity (virus, insecticides) you would be studying the interaction between virus/insecticide and mosquito, not the disease. Studies of mosquito-borne diseases would benefit of the availability of methods

that allow large-scale functional cell-based screens, for example genome-scale screens for virus, bacteria or parasite entry, innate immunity, or resistance to insecticides and other toxins.

To clarify this point, we have now changed the text as follows: “Studies of mosquito-borne **pathogens** would benefit of the availability of methods that allow large-scale functional cell-based screens, for example genome-scale screens for virus, bacteria or parasite entry, innate immunity, or resistance to insecticides and other toxins.”

Reference your own work: Previously, we developed such a method in Drosophila cells

Accepted

4. “Strong”, not “strongly”: we selected a single, strongly mCherry-positive

Accepted

5. The info on which DmelU6 promoter was used is only present in the methods. I

would add in the main text that you used U6-2 as a starting point on the ortholog search. I also have a comment on this and it is to make sure that you used U6-2 (it is not used much in in vivo genetic engineering because it displays low expression of the gRNAs, whereas U6-1 and U6-3 are widely used as gRNA promoters). I think that, in a paper where you are trying to find highly expressed U6s, it is worth mentioning and discussing: we first used BLAST and multiple alignment to identify orthologs of the Drosophila U6 promoter

Although we began our BLAST search with Drosophila U6-2, we ended up finding all U6 snRNA orthologs in each mosquito species. Based on the results, we aligned the promoter sequences and continued our experiments with those containing the conserved PSEA and TATA had having evidence of expression. We now reword the paragraph as follows: “To choose promoters, we first used BLAST and multiple sequence alignment to identify orthologs of the Drosophila U6-2 (snRNA:U6:96Ab) promoter and chose eleven orthologous promoters from U6 snRNAs of Anopheles, Culex, or Aedes (Figure 3a). When possible, we selected a minimum of three promoters per species, prioritizing U6 promoters that contain an intact pol III bipartite promoter motif and for which RNA-seq data suggests they are expressed in cell lines and in adult tissues (see Methods).” (**Lines 162-168**)

Importantly, repeating the approach using U6-1 or U6-3 from Drosophila results in the same mosquito snRNAs.

6. VEuPathDB references only for AGAP13695, not for the rest of U6s: RNA-seq data suggests they are expressed in cell lines and in adult tissues (see Methods).

We don't totally understand the reviewer's concern, because all the genes listed appear in VEuPathDB, in the VectorBase section. For instance, please follow the link below to see the experimental RNAseq trace for AGAP013557:

https://vectorbase.org/vectorbase/app/jbrowse?loc=AgamP4_2L%3A44242151..44243167&data=%2Fvectorbase%2Fservice%2Fjbrowse%2Ftracks%2FagamPEST&tracks=gene%2CNRDB%20Protein%20Alignments%2Cest%2CagamPEST_SRP052079_SRP052081_SRP052083_SRP052084_SRP052085_SRP052092_SRP052093_SRP052097_SRP052102_SRP052109_SRP052110_SRP052112_SRP052143_SRP052167_ebi_rnaSeq_RSRC%201_Sua5B_cell_line_nonunique%20Coverage&highlight=

7. KO is used throughout the text and it has already been introduced previously as “knockout”, use “KO”: all mosquito promoters tested elicited measurable knockout

We now use KO throughout.

8. This refers to 1e, not 1d. Should be 1d given the flow of the text. overall mean KO efficiency of about 30% (Fig. 1d, Supplementary Figure 2b)

Accepted.

9. In this case 1d is correct but should be moved to 1e since the results are shown after the Culex ones. with about 27% mean KO efficiency (Fig. 1d, Supplementary Figure 2b).

Accepted.

10. No comma. allowing in some cases, inter-species

We missed a comma after “allowing”. We’ve now added it. (Line 117)

11. Where do these cells come from? To this point, there is no mention of their creation,
only Sua-5B-IE8. We used the Anopheles coluzzii Sua-5B- IE8-Act::Cas9-2A-Neo cell line

We now add a new **Figure 2** that illustrates more clearly the creation of the Sua-5B- IE8-Act::Cas9-2A-Neo cell line.

12. Citation #15 has no validity here and, unless I am wrong, the data from 9 only elicited PTP-ER as a factor. If the authors write that PTP-ER (and the same occurs for the other genes) is a negative regulator of MAPK, which in turn can be suppressed with trametinib, both claims should be backed up with references. This, as mentioned, is a common problem of the paper. We first chose five genes that had

previously been shown to be drug-resistance factors in Drosophila cells^{9,15}

This is our mistake. We have removed this reference.

14. You did not perform CRISPRa or RNA knockdown screens in the paper. This includes not only application of CRISPR-Cas9 knockout screening but also CRISPR/Cas-based

activation and RNA knockdown screens.

We have removed reference to these future applications of RMCE screens in mosquito cells.

This is almost copy-pasted from paragraph 2 (see point 3 above). CRISPR pooled screening in these cells can be combined with a variety of cell-based assays, including assays relevant to virus or parasite entry, innate immunity, resistance to insecticides and other toxins

We have now significantly expanded and modified the introduction and discussion, and this statement has been changed.

Reviewers' Comments:

Reviewer #1:

Remarks to the Author:

The authors have put a large amount of effort into addressing the reviewers' comments with extensive rewrites and new data. I am satisfied that my comments have been sufficiently addressed.

Reviewer #2:

Remarks to the Author:

Dear editors and authors,

I think the manuscript is much improved and am happy that the authors have satisfactorily responded to my queries. I have a few minor points below, mostly about writing structure and nomenclature, but I am content with the much-improved manuscript. I think that the science presented in the paper represents an advancement of the field. Well done to the authors.

Important points

Figure 3 – The headers of each of the graphs should be aligned (d,e) and not get into another panel (3c – Sua looks like part of the 3a panel).

Line 185 – Figure 3c, not 1c

Line 192 – Figure 3d, not 1d

Line 199 – Figure 3e, not 1e

Line 224 – Do the authors mean 3c? That's the Sua-5B Anopheles cell line, not 3d as mentioned in the manuscript

Minor points

Line 25: "cultured mosquito cells" instead of "mosquito-cultured cells".

Line 29: screening of mosquito cell lines

Line 30-33: Change "Engineered and made available mosquito cell lines modified to express Cas9" to "engineered mosquito cell lines to express Cas9" – modified in this case is redundant as engineering means modifying. The whole sentence reads weird.

Line 44-46 and 71-73: There is a difference between virus and disease, and both times the authors have them partially wrong: Viruses are not diseases in the first instance (West Nile Virus), and diseases are not viruses in the second (encephalitis, fever). West Nile virus is the etiological agent that causes West Nile fever, for example.

Line 51-53: I assume this is talking about the interactions you may find between pathogen and mosquito host, not mammals. Please make it clear in here, mainly because the next paragraph starts with the same sentence for mammalian cells and how key insights have already been found.

Line 240: we used the Anopheles coluzzi Sua-5B-IE8-Act:Cas9-2A-Neo... it is redundant as all has been done using this line and it comes after a cell line-heavy paragraph, so the intra-sectional flow can be improved.

Line 266: "screen-ready" – you called it "CRISPR-ready" earlier. Is that just because of the two different applications? If so, you still use CRISPR in the screen, so it is still CRISPR-ready in essence.

Line 293: In a pilot pooled CRISPR KO... Fix that sentence, it does not read properly.

Line 313: polIII promoter inconsistent with the rest of the manuscript

Line 321: Interestingly, to our knowledge, we report...

Line 415: Double space after described?

Line 464: Figure 3a, not Fig.3a, same with the supplementary figure name

Line 512: KO

As a minor personal note, I would suggest that the authors abbreviate the name "Sua-5B-IE8-Act:Cas9-2A-Neo" cell line to something more legible.

As always, happy to respond to correspondence regarding this review at gkt5113@psu.edu.
Gerard

Dear Editors,

We have now addressed the points raised by the reviewers. Our points are in blue.

Sincerely,
Authors

REVIEWERS' COMMENTS

Reviewer #1 (Remarks to the Author):

The authors have put a large amount of effort into addressing the reviewers' comments with extensive rewrites and new data. I am satisfied that my comments have been sufficiently addressed.

We thank the reviewer for their comments that helped us improve the manuscript.

Reviewer #2 (Remarks to the Author):

Dear editors and authors,

I think the manuscript is much improved and am happy that the authors have satisfactorily responded to my queries. I have a few minor points below, mostly about writing structure and nomenclature, but I am content with the much-improved manuscript. I think that the science presented in the paper represents an advancement of the field. Well done to the authors.

We thank the reviewer for these positive comments and the for the previous comments and suggestions that improved the manuscript.

Important points

Figure 3 – The headers of each of the graphs should be aligned (d,e) and not get into another panel (3c – Sua looks like part of the 3a panel).

Line 185 – Figure 3c, not 1c

Line 192 – Figure 3d, not 1d

Line 199 – Figure 3e, not 1e

Line 224 – Do the authors mean 3c? That's the Sua-5B Anopheles cell line, not 3d as mentioned in the manuscript

Thank you for your attention to this. We have now made the change.

Minor points

Line 25: “cultured mosquito cells” instead of “mosquito-cultured cells”.

Line 29: screening of mosquito cell lines

Line 30-33: Change “Engineered and made available mosquito cell lines modified to express Cas9” to “engineered mosquito cell lines to express Cas9” – modified in this case is redundant as engineering means modifying. The whole sentence reads weird.

Line 44-46 and 71-73: There is a difference between virus and disease, and both times the authors have

them partially wrong: Viruses are not diseases in the first instance (West Nile Virus), and diseases are not viruses in the second (encephalitis, fever). West Nile virus is the etiological agent that causes West Nile fever, for example.

Line 51-53: I assume this is talking about the interactions you may find between pathogen and mosquito host, not mammals. Please make it clear in here, mainly because the next paragraph starts with the same sentence for mammalian cells and how key insights have already been found.

Line 240: we used the Anopheles coluzzi Sua-5B-IE8-Act:Cas9-2A-Neo... it is redundant as all has been done using this line and it comes after a cell line-heavy paragraph, so the intra-sectional flow can be improved.

Line 266: “screen-ready” – you called it “CRISPR-ready” earlier. Is that just because of the two different applications? If so, you still use CRISPR in the screen, so it is still CRISPR-ready in essence.

Line 293: In a pilot pooled CRISPR KO... Fix that sentence, it does not read properly.

Line 313: polIII promoter inconsistent with the rest of the manuscript

Line 321: Interestingly, to our knowledge, we report...

Line 415: Double space after described?

Line 464: Figure 3a, not Fig.3a, same with the supplementary figure name

Line 512: KO

We have made all of these changes. Thank you for pointing out.

As a minor personal note, I would suggest that the authors abbreviate the name “Sua-5B-IE8-Act:Cas9-2A-Neo” cell line to something more legible.

We apologize that we will not make this change because the cells are too far along the process of being distributed.

As always, happy to respond to correspondence regarding this review at gkt5113@psu.edu.

Gerard